# Phylogenomic and mitogenomic data can accelerate inventorying of tropical beetles during the current biodiversity crisis

**Michal Motyka†, Dominik Kusy†, Matej Bocek, Renata Bilkova, Ladislav Bocak\***

Czech Advanced Science and Technology Institute, Palacký University, Olomouc, Czech Republic

**Abstract** Conservation efforts must be evidence-based, so rapid and economically feasible methods should be used to quantify diversity and distribution patterns. We have attempted to overcome current impediments to the gathering of biodiversity data by using integrative phylogenomic and three mtDNA fragment analyses. As a model, we sequenced the Metriorrhynchini beetle fauna, sampled from ~700 localities in three continents. The species-rich dataset included ~6500 terminals, ~ 1850 putative species delimited at 5% uncorrected pairwise threshold, possibly ~1000 of them unknown to science. Neither type of data could alone answer our questions on biodiversity and phylogeny. The phylogenomic backbone enabled the integrative delimitation of robustly defined natural genus-group units that will inform future research. Using constrained mtDNA analysis, we identified the spatial structure of species diversity, very high species-level endemism, and a biodiversity hotspot in New Guinea. We suggest that focused field research and subsequent laboratory and bioinformatic workflow steps would substantially accelerate the inventorying of any hyperdiverse tropical group with several thousand species. The outcome would be a scaffold for the incorporation of further data from environmental sequencing and ecological studies. The database of sequences could set a benchmark for the spatiotemporal evaluation of biodiversity, would support evidence-based conservation planning, and would provide a robust framework for systematic, biogeographic, and evolutionary studies.

**\*For correspondence:**
ladislav.bocak@upol.cz

†These authors contributed equally to this work

**Competing interest:** The authors declare that no competing interests exist.

## Editor's evaluation

This manuscript provides clear ideas regarding the usage of next-generation sequencing data, and of more traditional mtDNA markers, to rapidly increase biodiversity inventories. You demonstrate how biodiversity information analyses done in the Metriorrhynchini, a hyperdiverse tropical insect group, can be rapidly expanded via targeted field research and large-scale sequencing. The study sets a benchmark for the spatiotemporal evaluation of tropical biodiversity, supports evidence-based conservation planning, and provides a robust framework for systematic, biogeographic, and evolutionary studies.

## Introduction

The number of known insects surpasses that of all other terrestrial groups (*Mora et al., 2011*), and we need much more detailed information to fully understand their diversity. Currently, the available biodiversity data are far from complete, and the majority of insect species remain undescribed (*Novotny et al., 2006*; *Srivathsan et al., 2019*). In addition, robust phylogenetic hypotheses are lacking for

most lineages, and the genera and tribes are often artificial assemblages which are not relevant to evolutionary and biodiversity research. Therefore, we need to gather new information in order to advance our understanding of evolutionary and genetic relationships, and to build a phylogenetic scaffold for comprehensive taxonomic, biogeographic, and evolutionary studies that would be indispensable for biodiversity management.

Descriptive, morphology-based insect systematics is not keeping pace with the rapid loss and degradation of natural habitats (*Theng et al., 2020*), and with the ongoing decline in insect abundance as a result of human activities and climate change (*van Klink et al., 2020*). The largest taxonomic journal, *Zootaxa*, published almost 30,000 studies describing >60,000 new species and these represent over a quarter of all new animal species reported in 2001–2020 (*Zhang, 2021*). Although these numbers are impressive, they also show how labor-intensive is taxonomic research if, in average, only two new species are reported in a publication. To accelerate the cataloguing of biodiversity, it is vital to gather new material suitable for molecular analyses and to combine available molecular methods with traditional approaches (*Riedel et al., 2013*; *Srivathsan et al., 2019*; *Yeo et al., 2020*; *Sharkey et al., 2021*). DNA data are indisputably a valuable source for modern biodiversity research, and they can address both shallow and deep relationships (*Tautz et al., 2003*; *Hajibabaei et al., 2007*). There are two principal sources of short-fragment data: voucher-based DNA sequences typically produced by systematists (*Riedel et al., 2013*; *Yeo et al., 2020*; *Sharkey et al., 2021*), and DNA sequences produced by an ecosystem-based sequencing that does not associate individual samples with Linnean names (*Andújar et al., 2015*; *Srivathsan et al., 2019*). It is the responsibility of systematic biologists to assemble the natural system, that is, we need to reliably delimit genus- and tribe-level taxa, to make their ecological and distribution attributes informative. But the short fragments are often unsuitable for the building of deep phylogenies and large genomic datasets need to be used (*McKenna et al., 2019*; *Baca et al., 2021*). These include transcriptomes, whole genome

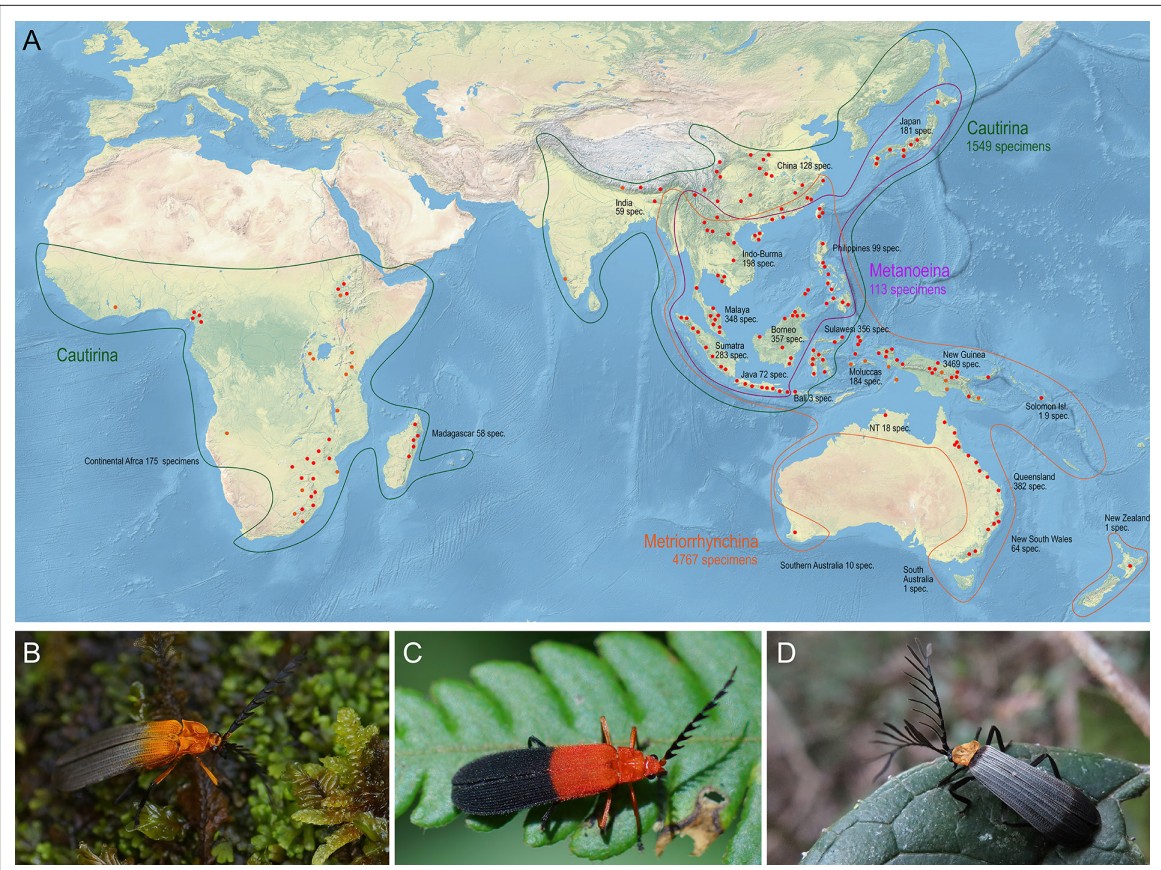

**Figure 1.** Distribution and appearance of metriorrhynchine net-winged beetles. (**A**) Distribution of Metriorrhynchini with major sampled localities designated by red dots. The numbers of analyzed specimens from individual regions are shown for regions and subtribes. (**B–D**) – General appearance of Metriorrhynchini.

sequences and anchored hybrid capture datasets. Using all information, a robust and stable natural classification will significantly facilitate detailed research into the spatial and temporal distribution of biodiversity (*Morrison et al., 2009*; *Thomson et al., 2018*). As an ultimate goal, we should attempt to construct a complete tree of life, or at least its backbone, which is invaluable in aiding the selection of groups for more detailed analyses (*Chesters, 2017*; *McKenna et al., 2019*). With a well-defined high-level classification, it is paramount to exploit all accessible data. We assume that voucher-based molecular phylogenies provide much-needed tools to researchers working on site-based biodiversity assessments (*Andújar et al., 2015*; *Srivathsan et al., 2019*) and that, in turn, the data produced by environmental and ecosystem-focused sequencing contribute to building the tree-of-life (*Arribas et al., 2016*; *Bocak et al., 2016*).

We have used hyperdiverse tropical metriorrhynchine beetles (Coleoptera, Lycidae, Metriorrhynchini) as our model. This net-winged beetle tribe contains >1500 recognised species, mostly found in the Old-World tropics (*Figure 1A*), and their classification is complicated by the complex taxonomic history (*Bocak et al., 2020*; see Appendix introductory information). The phenetic plasticity of Metriorrhynchini is relatively high (*Figure 1B–D*), but many distant species resemble each other due to convergent selection in Mullerian rings (*Bocek et al., 2019*; *Motyka et al., 2020*; *Motyka et al., 2021*). Therefore, unrelated taxa were often assumed to be closely related due to misleading morphological similarities. Although there are over 40 genera in the tribe, three-quarters of the species have been described in five ambiguously defined genera (*Xylobanus*, *Cautires*, *Trichalus*, *Metriorrhynchus*, and *Cladophorus*). Sometimes a single genus contains species from different subtribes (*Bocak et al., 2020*). In this respect, the Metriorrhynchini is a typical species-rich tropical insect group without well-founded classification and the paucity and inaccuracy of available data (*Letsch et al., 2020*). As a result, unlike vertebrates, these poorly known insect groups have not been considered for use in large-scale, integrative projects and data metanalyses (*Myers et al., 2000*; *Holt et al., 2013*) and have contributed little to our understanding of global biodiversity patterns.

The principal objective of this study is to demonstrate how biodiversity information for a hyperdiverse tropical group can be rapidly expanded via targeted field research and large-scale sequencing of transcriptomes, genomes, and short mtDNA fragments. Our investigation comprised four distinct steps. First, we assembled material from several hundred localities on three continents (*Figure 1*, *Table 1*). Second, as hyperdiverse groups are difficult to tackle and the current classification is unreliable, we attempted to compartmentalise diversity using phylogenomics. We then produced a tree,

**Table 1.** The numbers of sampled localities per region.
Details in *Appendix 1—table 1*.

| Area | Localities | Area | Localities |
|---|---|---|---|
| Australian region | 298 | Sino-Jap. region | 79 |
| Australia | 118 | China | 51 |
| New Guinea & Solomons | 179 | Japan | 28 |
| New Zealand | 1 | | |
| Wallacea | 49 | | |
| Moluccas | 15 | Oriental region | 206 |
| Sulawesi | 34 | S.India & Ceylon | 3 |
| | | E.India & Burma | 12 |
| Afrotropical Region | 64 | E.Indo-Burma | 44 |
| West Africa | 1 | Malay Peninsula | 57 |
| Guinean Gulf | 11 | Sumatra | 23 |
| Ethiopia | 6 | Java & Bali | 15 |
| East Africa | 10 | Philippines | 33 |
| South Africa | 25 | | |
| Madagascar | 11 | Total | 696 |

using all available data, to estimate species limits, intraspecific genetic variability, and species ranges. Finally, the tree was pruned and used to estimate shallow phylogenetic relationships, total and regional species diversity, and endemicity, and to define generic ranges and continental-scale range shifts. The applied methods of transcriptome and mtDNA analyses are widely used. The genomic datasets dominate among works focusing on deep relationships (transcriptomes and anchored hybrid capture; *Misof et al., 2014*; *Kusy et al., 2019*; *McKenna et al., 2019*; *Baca et al., 2021*). The mitochondrial markers have been used mainly to study the phylogeny of restricted clades (*Toussaint et al., 2014*; *Sharkey et al., 2021*). Until now, the genomic and mitochondrial data have seldom been combined to get simultaneously the phylogenetic backbone for the mid-rank classification (subtribes, groups of genera, generic limits) and the estimations of species diversity (e.g. *Talavera et al., 2021*). Our information and phylogenetic hypotheses can be a resource for higher level phylogenetics, population genetics, phylogeographic studies, and biodiversity estimation. At the same time, we want to show how limited our taxonomical knowledge is and how this lack is hindering biodiversity research and management (*Thomson et al., 2018*).

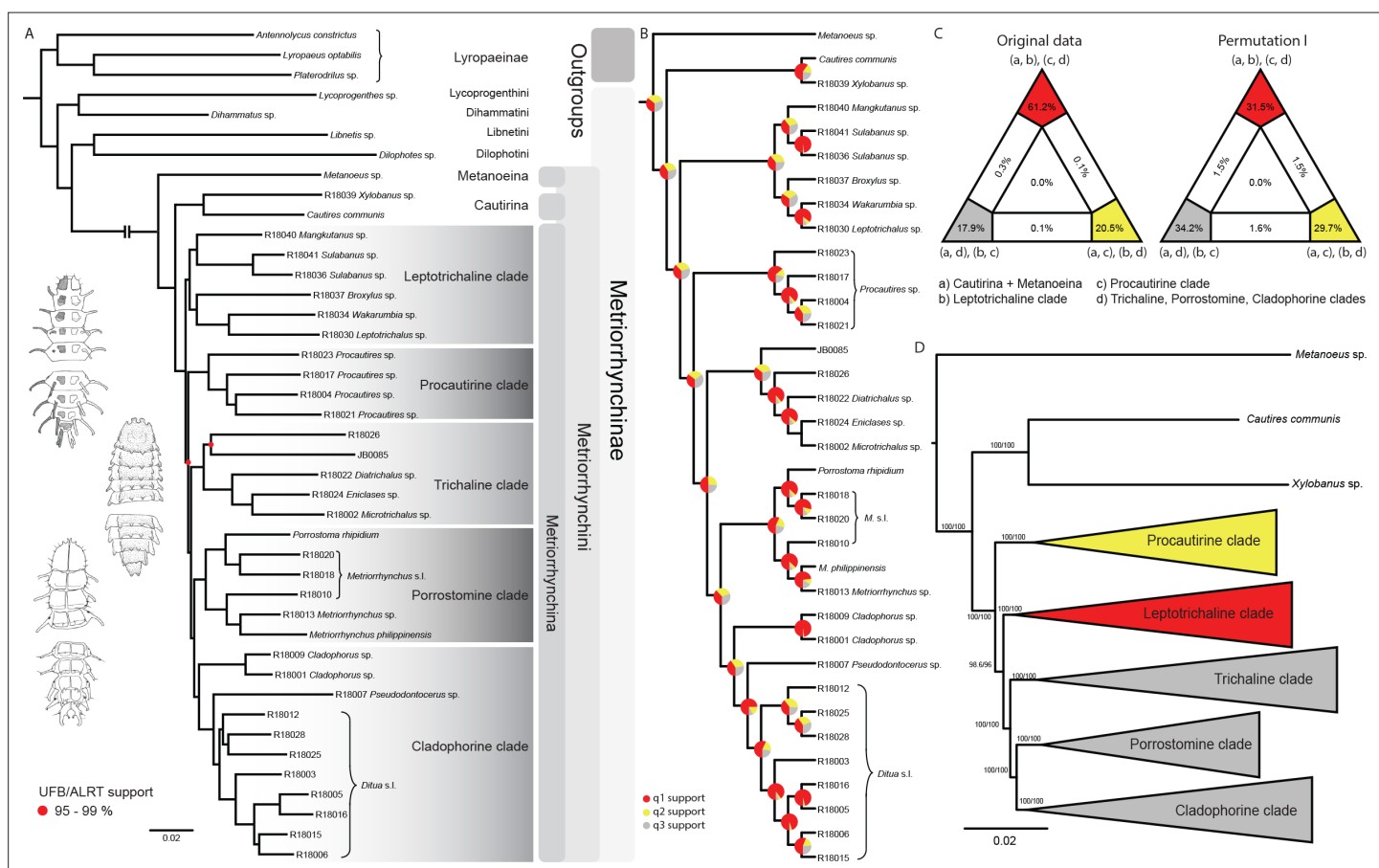

**Figure 2.** Topologies recovered by phylogenomic analyses. (**A**) Phylogenetic relationships of Metriorrhynchinae based on the ML analyses of the concatenated amino-acid sequence data of supermatrix F-1490-AA-Bacoca-decisive. Unmarked branches are supported by 100/100 UFB/alrt; red circles depict lower phylogenetic branch support. (**B**) Phylogenetic relationships of Metriorrhynchini recovered by the coalescent phylogenetic analysis with ASTRAL when analysing the full set of gene trees (4109 gene trees inferred at the nucleotide level). Pie charts on branches show ASTRAL quartet support (quartet-based frequencies of alternative quadripartition topologies around a given internode). Outgroups taxa are not shown. (**C**) Results of FcLM analyses for selected phylogenetic hypotheses applied at the amino-acid sequence level (supermatrix F). (**D**) Alternative phylogenetic relationships of Metriorrhynchinae based on the ML analyses of the concatenated amino-acid sequence data of supermatrix A-4109-AA. Numbers depict phylogenetic branch support values based on 5000 ultrafast bootstrap replicates.

## Results

### Sampling of the Metriorrhynchini range

In total, we monitored almost 800 localities, 696 of them with occurrences of the Metriorrhynchini (Tabs. 1, *Appendix 1—table 1*). The distribution of sampling sites was partly biased due to the large extent of the Metriorrhynchini range, limited time and funds, different goals of various expeditions, and logistic problems (inaccessible regions, legal obstacles). The densest sampling is available from the Sundaland and New Guinea, while India and the Afrotropical region are under-sampled (*Figure 1A*).

### Assembly of the phylogenomic tree

The phylogenomic dataset contained 35 Metriorrhynchini terminals (*Appendix 1—table 2*), seven outgroups, and ~4200 orthologs (1.9–5.7 × 10⁶ aligned positions; *Supplementary file 1*;

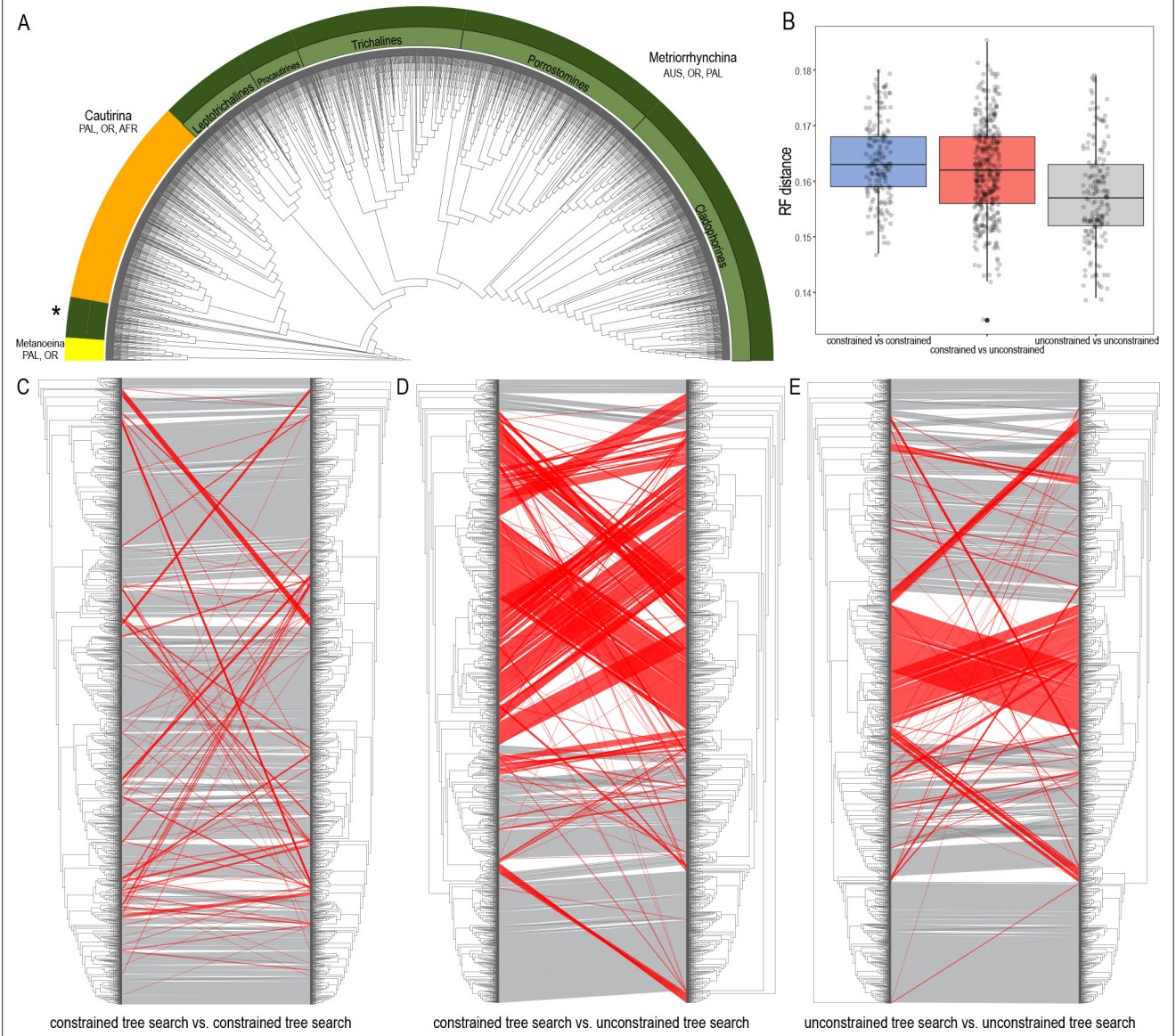

**Figure 3.** Topologies recovered by mitogenomic analyses. (**A**) Relationships of Metriorrhynchini recovered by the constrained analysis of the pruned dataset at 2% distance. (The full resolution tree is shown in *Source data 2* along with a tree recovered from the analysis of a complete dataset of 6429 terminals in *Source data 1*), asterisk designates a grade of Metriorrhynchina-like taxa found in a position in conflict with their morphology; (**B**) A chart of Robinson-Foulds distances among topologies inferred by repeated runs of the constrained and unconstrained analyses; (**C**) A comparison of the results obtained by two runs of the constrained analysis; (**D**) A comparison of trees inferred with/without the phylogenomic backbone; (**E**) A comparison of results obtained by two runs of the unconstrained analysis. The red lines designate terminals with conflicting positions in compared trees.

*Appendix 1—table 3*; *Appendix 1—table 4*). The tree shown in *Figure 2A* was produced using maximum likelihood (ML) analyses, whereas the coalescent method produced the topology shown in *Figure 2B*; additional trees are shown in *Appendix 1—figures 1–8*. For details on the data sets' characteristics see *Appendix 1—figures 9–12*. Phylogenomic analyses resolved three subtribes (Metanoeina [Metriorrhynchina, Cautirina]), and five clades were regularly recovered within the Metriorrhynchina, that is, the hereby defined procautirines, leptotrichalines, trichalines, porrostomines, and cladophorines. Different settings (see Materials and methods) produced slightly different topologies and shifted the positions of the leptotrichalines and procautirines (*Figure 3D*). However, the monophyly of major subclades was not affected. The FcLM analysis favored a deeper position for the leptotrichaline clade (61.2%; *Figure 2C*, *Appendix 1—figure 13*). The position of the remaining terminals was stable across all analyses. All phylogenomic analyses question the definitions of some species rich genera (*Appendix 1—figures 1–8*) that are either polyphyletic (e.g. *Cladophorus*; 131 described species, most of them recovered in the *Ditua* subclade) or paraphyletic (*Metriorrhynchus* as a grade and *Porrostoma* in the terminal position; 194 species, see *Figure 2A and B*, *Appendix 1—figures 1–8*).

## Constrained mitogenomics

The mtDNA database contained >11,500 mtDNA fragments (5935 *cox1*, 2381 *rrnL*, and 3205 *nad5*) representing 6429 terminals (2930 aligned positions). Using these data, we inferred additional trees using the constrained positions of 35 terminals whose relationships were determined through

**Table 2.** The numbers of described species and identified mOTUs (molecular operational taxonomic units) at 2% and 5% thresholds per region and the total number of OTUs identified for subtribes. Based on morphological identification, the OTUs of the kassemiine and other deeply rooted clades are included in Metriorrhynchina.

| Region | Metriorrhynchina described/ analyzed at 2%/5% | Cautirina described/ analyzed at 2%/5% | Metanoeina described/ analyzed at 2%/5% | Metriorrhynchini described/ analyzed at 2%/5% | Ratio Analyzed/ described |
|---|---|---|---|---|---|
| Australian region | 639/1608/1239 | | | 639/1608/1239 | 2.52–1.93 |
| Australia | 196/167/133 | | | 196/167/131 | 0.85–0.67 |
| New Guinea | 423/1434/1105 | | | 423/1434/1105 | 3.39–2.61 |
| Solomon Isl. | 21/9/9 | | | 21/9/9 | 0.43 |
| Wallacea | 162/174/162 | 14/10/9 | | 176/184/171 | 1.05–0.97 |
| Philippines | 51/18/18 | 45/12/12 | 8/3/3 | 104/33/33 | 0.32 |
| Continental Asia | 43/52/42 | 331/330/257 | 30/34/31 | 404/416/330 | 1.03–0.82 |
| Sundaland | 36/44/39 | 201/184/146 | 24/19/17 | 261/247/202 | 0.95–0.77 |
| Indo-Burma | 6/7/7 | 62/52/42 | 3/4/4 | 74/63/53 | 0.85–0.72 |
| China, Japan | 1/1/1 | 53/75/58 | 1/11/11 | 55/87/70 | 1.58–1.27 |
| India | | 35/19/18 | 2/0/0 | 37/19/18 | 0.51–0.49 |
| Afrotropical region | | 231/104/94 | | 231/104/94 | 0.46–0.41 |
| Sub-Saharan Africa | | 178/74/65 | | 178/74/65 | 0.42–0.37 |
| Madagascar | | 53/30/29 | | 53/30/29 | 0.57 |
| Total number of OTUs | 895/1852/1445 | 641/456/369 | 38/37/34 | 1574/2345/1848 | 1.50–1.17 |

phylogenomic analyses, and the free positions of the other ~6400 terminals (*Source data 1*). The units based on uncorrected pairwise distances represent molecular operational taxonomic units (mOTUs), considered to be putative species, or 'species' for short. Depending on the applied 2% and 5% thresholds, we identified 34–37 mOTUs in the Metanoeina clade and 369–456 mOTUs in Cautirina. The major Metriorrhynchina clade (1376–1763 mOTUs) included procautirines, leptotrichalines, trichalines, porrostomines, and cladophorines. In addition, we identified several deeply rooted lineages, the kassemiines, and another five small clades (69–89 mOTUs in total; *Source data 2*), each of which comprised a limited number of species. As phylogenomic data for these terminals are still lacking, their positions were determined based only on mtDNA data and they are included in Metriorrhynchina, based on morphological traits (*Table 2*). The number of mOTUs does not include ~50 mOTUs for which *cox1* was unavailable.

## Pruned mitogenomic tree with and without constraints

The dataset was subsequently pruned to a single terminal per mOTU based on 2% and 5% distance (see below) and was analyzed both with and without topological constraints (*Figure 3A*; *Source data 2 and 3* show the pruned trees at 2% levels to capture the intraspecific genetic variability within the clusters of closely related mOTUs). Repeated runs with different starting seeds identified terminals with unstable positions (*Figure 3A–C*). The major clades were generally stable, whereas small, deeply rooted clades were prone to 'wandering' around the tree, as were distinct singletons. The trees that resulted from each of the seed-specific 19 ML runs differed slightly; tree similarity was thus evaluated using the Robinson-Foulds index, with values ranging from 0.180 (most similar) to 0.147 (most distant; *Appendix 1—table 4*).

## Tree congruence

The degree of incongruence between selected topologies is shown in *Figure 3C–D*. The unconstrained analysis of mitochondrial data yielded a topology with a high number of terminals that were recovered in positions incongruent with their morphology (*Figure 3D and E*, *Source data 3*). The same dataset, when analyzed using the constrained position of 35 terminals (based on their relative relationships inferred by prior phylogenomic analyses), produced a topology with a much lower proportion of terminals in dubious positions (*Figure 3C*, *Source data 2*). The composition of the constituent clades is based on the topology recovered by the constrained mtDNA analyses and the position of genera was validated by morphological comparisons of vouchers with the type species of earlier described genera (all redescribed by *Bocak, 2002*). The named genera assigned into individual subclades are shown individual clades are characterized in Appendix results.

## Species diversity

To investigate the total and regional species diversity of the Metriorrhynchini, we analyzed a dataset comprising 5935 of the 6429 terminals for which the *cox1* mtDNA fragment was available (*Figure 3A*; *Supplementary file 1*). For the Metriorrhynchini, we identified 1848 and 2345 mOTUs using the 5% and 2% thresholds, respectively (*Appendix 1—figure 14*). We disregarded the presence of ~50 mOTUs (494 terminals) for which *cox1* was missing. The number of mOTUs based on the *cox1* analysis varied by thresholds and the number of delimited OTUs increased relatively slowly with decreasing threshold values from 1% to 10% (*Appendix 1—figure 14*).

Using an earlier published literature review (*Bocak et al., 2020*), we updated species lists for the Cautirina (641 spp. described species), Metanoeina (38 spp.), and Metriorrhynchina by adding taxa described in 2020 and 2021. By analysing DNA data, we identified 34–37 putative spp. of Metanoeina, 369–456 spp. of Cautirina, and 1445–1852 spp. of Metriorrhynchina, depending on the applied mtDNA uncorrected pairwise 5% and 2% mtDNA distance thresholds. The numbers of species per subregion, along with the estimated ratios between formally described and estimated species diversity, are shown in *Table 2* for the 2% and 5% threshold (further information in *Appendix 1—figure 14*). Using both thresholds, 2% and 5%, the numbers of putative species surpass the numbers of species reported in the literature.

We observed very high species turnover even if 5% threshold was applied for delimitation. Only four mOTUs have been recorded in two landmasses separated by a deep-sea (> 200 m). The faunas of Sulawesi and the islands across Wallace's and Weber's lines share two mOTUs, one mOTU was

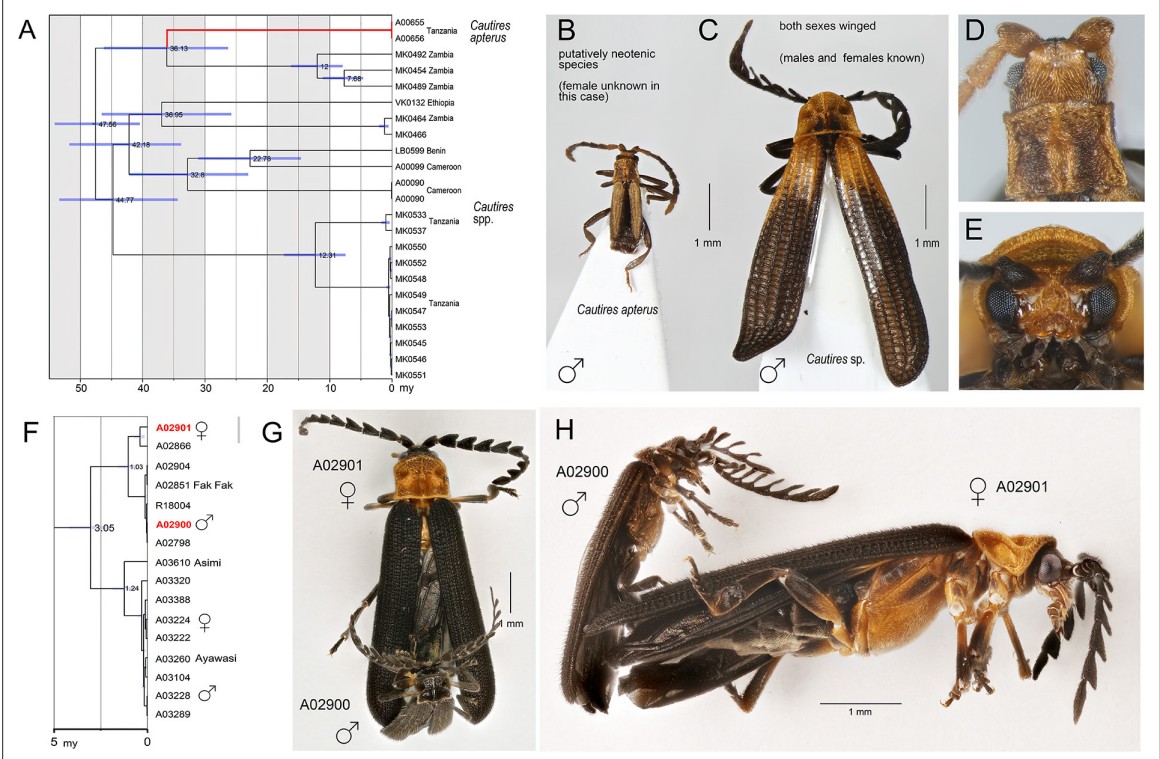

**Figure 4.** Identification of sexual dimorphism by large-scale biodiversity inventory. (**A**) Relationships of lineages with modified ontogeny, the dated tree; (**B, D**) General appearance and head of *Cautires apterus*, a putative neotenic species; (**C, E**) ditto of the close relative with both sexes winged. Mimetic sexual dimorphism identified during diversity survey. (**F**) The dated tree, red colored terminal labels designate the individuals shown in G and H; (**G**) Dorsal view of individuals in copula; (**H**) Ditto, lateral view. Except of collecting individuals in copula, DNA-based assessment of relationships is the only option as the species are sexually dimorphic and no morphological traits indicate their conspecifity.

simultaneously identified in Laos and Luzon and one species in New Guinea and the Solomon islands. Similarly, only sixteen mOTUs were distributed across two landmasses separated by an inundated shelf (sea depth <100 m). Nine mOTUs were distributed in two or more islands of Southeast Asia and seven species were found in both New Guinea and Australia. The centres of species diversity of the Metriorrhynchini are New Guinea (1,105 putative spp. at 5% threshold) and the seasonally to perennially humid areas of the Sundaland (202 spp. at 5% threshold). The results suggest substantial modifications to the generic limits and ranges for numerous taxa that had been previously delimited (*Appendix 1—figure 15*).

Having the extensive the mtDNA topologies, we looked for examples of the evolution of neoteny and mimetic polymorphism. The detailed inspection of trees identifies the closest available relative of a putative neotenic, *Cautires apterus* (Cautirina). This species is morphologically very distinct (*Figure 4B and D*). The dated subtree indicates the recent origin of morphological divergence (*Figure 4A*). The mtDNA analyses recover some species with pronounced sexual dimorphism, such as an unidentified genus and species of the procautirine clade (*Figure 4F–H*). The origin of the polymorphism is putatively very recent (*Figure 4F*).

## Discussion

In the context of the present loss of biodiversity (*Sodhi et al., 2004*; *Hallmann et al., 2017*; *Theng et al., 2020*), large-scale genomic resources are urgently needed for biodiversity assessment and conservation (*Hajibabaei et al., 2007*; *Krehenwinkel et al., 2019*). Molecular data cannot replace morphology-based taxonomy (*Figure 3C–D*; *Thomson et al., 2018*), but the analyses of our dataset complement and facilitate traditional biodiversity research in several directions. Our first step is to compartmentalize hyperdiverse Metriorrhynchini into manageable natural units (*Figure 2*). The densely sampled phylogeny identifies tribal and generic limits. It provides a useful foundation for

detailed taxonomic research through the identification of weak areas in earlier classifications and points out the clades with undescribed diversity and non-monophyletic genera (e.g. several hundred species of *Ditua* and paraphyletic *Metriorrhynchus*; *Figure 3*, *Appendix 1—figure 15*). Furthermore, the analyses of species-rich datasets identify the areas with high species diversity as one of the critical conservation value parameters (*Table 2*; *Baselga, 2010*; *Srivathsan et al., 2019*). Traditional taxonomic research costs time and money, and the number of newly described species is relatively low if we consider the enormous diversity of tropical insects (*Novotny et al., 2006*; *Sangster and Luksenburg, 2015*). Therefore, we use DNA-based units as a provisional descriptor of species diversity (*Hebert et al., 2003*; *Monaghan et al., 2009*), and subsequently as a source for integrative taxonomy (*Source data 1–3*; *Srivathsan et al., 2019*). The presented large-scale monitoring project provides information on relationships (*Figures 2 and 3*), genetic divergence (*Source data 1–3*), turnover (*Table 2*), the extent of generic and species ranges (*Appendix 1—figure 15*, *Source data 1–3*), and on evolutionary phenomena that are usually studied using a few model organisms (*Figure 4*). Using phylogenomics and voucher-based sequencing, we show that taxonomic literature has provided insufficient and sometimes erroneous information, even after the formal consolidation of scattered descriptions (*Bocak et al., 2020*). We show that a taxon-focused continental scale project can effectively assemble comprehensive data for diversity of tropical insects.

## Continent-wide taxon-specific monitoring of biodiversity: feasibility and impediments

Tissue and DNA archives have become critical in the assessment of biodiversity status (*Hajibabaei et al., 2007*; *Blom, 2021*). Although museomics is a potentially valuable source of data (*Gauthier et al., 2020*), in our case, museum collections are insufficient for filling data gaps due to the scarcity of material. For example, the Metriorrhynchini collection deposited in the Natural History Museum in London contains <3000 specimens, whereas there are ~6500 terminals in our dataset. At the beginning of our study, we faced critical absence of primary data. Therefore, we conducted intensive field research to obtain samples for a realistic assessment of the extant Metriorrhynchini diversity. We processed samples from our expeditions (most of which were focused on a range of topics over two decades between 2001 and 2019) and samples obtained through extensive collaboration with other researchers, both local and visiting, and with local naturalists whose contribution has increased with the growing number of citizen science projects (*Jaskula et al., 2021*; *MacPhail and Colla, 2020*). In such a way, we assembled a Metriorrhynchini tissue collection from almost 700 localities in three continents (*Table 1*, *Figure 1*). For several reasons our sampling is partly biased. We noted the serious loss of natural habitat in many regions. Previously described species were often collected in vicinity of seaports, but the lowland ecosystems are rapidly disappearing due to human exploitation. Therefore, type localities of many described species could not be sampled during recent expeditions and species known from museum collections are missing in our DNA dataset (*Jiruskova et al., 2019*). The habitat loss in South East Asia also affects other animal groups, and lowland primary forests are seriously endangered in the whole region (*Sodhi et al., 2004*; *Theng et al., 2020*). Further sampling bias is a consequence of the unsafe conditions and logistic problems in large areas of West Africa, Sahel, and the Congo Basin (*Figure 1A*), where net-winged beetles have not been systematically studied since the 1930 s. Additional data gaps are caused by strict biodiversity research restrictions (*Prathapan, 2018*). Regardless of these limitations, we believe that the assembled dataset is a foundation for a robust classification framework and a soundly based assessment of biodiversity. Our results show the importance of field research for biodiversity studies and systematics (*Basset and Lamarre, 2019*).

## Phylogenetic relationships: a scaffold for targeted research

Unresolved taxonomy is a common reason for the exclusion of specific groups from biodiversity research projects and this omission has an effect on conservation policies (*Gutiérrez and Helgen, 2013*). The current phylogenomic and mitogenomic phylogenetic hypotheses (*Figures 2 and 3*; *Source data 1–3*) supersede the morphology-based topologies (*Bocak, 2002*). The phylogenomic analysis incorporates a large amount of information, and we favor this method over morphological traits and short DNA sequences, both of which contain uncertainties (*McKenna et al., 2019*). Phylogenomics has resolved subtribe relationships and their internal structures. The analyzed 35 transcriptomes and low-coverage genomes were sufficient to identify five major Metriorrhynchina clades with

several hundred putative species each and also to identify the limits of genera, which can be tested using traditional taxonomic methods (*Figures 2 and 3A*; *Source data 1–3*).

The sampling strategy is critical for building a phylogenomic backbone. Our goal was to cover as many deep lineages as possible and simultaneously to limit the number of sequenced RNA samples to avoid high costs. Therefore, we sequenced RNAlater preserved tissues and conspecific vouchers prior to assigning tissue samples for transcriptomic analyses. In this way, two rounds of sequencing provided us with critical information based on evenly distributed anchor taxa. In the next step, we re-analyzed the short-fragment dataset (*Supplementary file 1*) using constrained positions for taxa whose relationships had already been recovered through phylogenomics (*Figures 2 and 3*). A stabilized phylogenomic backbone is critical for inferring a robust topology because the large-scale analyses of short mtDNA fragments are sensitive, even to the application of starting seeds, and they often produce topologies incongruent with morphological traits (*Figure 3B and E*; *Sklenarova et al., 2014*). Only several small lineages have remained unanchored by genomic data, owing to a lack of properly fixed samples (*Source data 2 and 3*). For example, four small clades are much more deeply rooted than their morphology suggests (*Figure 3A*, *Source data 2*) and additional data are needed to place them in a phylogenetic context. Despite some contentious relationships that need further investigation, 35 genomic samples, that is under 2% of species, sufficiently supported relationships among most terminal clades that approximately represent genera, groups of genera and subtribes. *Talavera et al., 2021* have shown that only six nuclear markers for 5%–10% of terminals can similarly stabilize the phylogeny of a species rich model group. We assume, that the combination of genomic and short DNA data can be valuable for building of the species-level trees of life.

We identified a substantial conflict between phylogenomic analyses, morphology-based classifications (*Bocak, 2002* and earlier studies cited therein), and the analyses of a few short DNA fragments (*Sklenarova et al., 2014*, Appendix introductory information). Our analyses confirm the monophyly of the recently described Cautirina and Metanoeina and redefined Metriorrhynchina except several unanchored lineages (*Sklenarova et al., 2013*; *Sklenarova et al., 2014*), but, for the first time, we can robustly recover subtribal relationships (*Figure 2A and B*). We reject most internal splits suggested by morphological and mtDNA analyses (*Bocak, 2002*; *Sklenarova et al., 2013*; *Sklenarova et al., 2014* and earlier studies cited therein). The present delimitation of five monophyla within Metriorrhynchina resolves the backbone of the subtribe that was contentious due to high levels of homoplasy in Sanger and morphology-based datasets. Similarly, some generic concepts are questioned as they have been mostly based on highly homoplastic traits (*Sklenarova et al., 2014*; *Kusy et al., 2019*). The taxonomic studies must consider the morphology along with molecular hypotheses. As morphology is not described in this study, we do not discuss the limits of individual genera and report only short information on newly defined subclades (Appendix results).

Our approach yielded a constrained phylogeny with 6429 terminals and almost 2000 mOTUs using 5% mtDNA distance threshold, and this provides the basis for the approximation of species diversity for constituent subclades and geographic regions (*Figure 3*; *Source data 2 and 3*). Concerning the extent of diversity, phylogenomic and mitochondrial data must be simultaneously analysed to provide a strong foundation for subsequent investigations (*Figure 3C*, *Talavera et al., 2021*). Phylogenomics cannot deal with thousands of species, and mitogenomic data are insufficient for the construction of robust deep relationships. The final steps are morphological validation of the proposed generic groups and genera (see Appendix results) and, in the future, formal descriptions of biodiversity using the Linnean classification. In such a way, the results of phylogenomic and mitogenomic inventory should be incorporated in the formal classification (*Godfray and Knapp, 2004*).

## Species diversity: literature data and reality

Here, we deal with a species rich tropical beetle tribe (> 1500 described species), and therefore we use the uncorrected pairwise distance thresholds for our diversity estimation (*Table 2*, *Appendix 1—figure 14*). The application of any threshold is a compromise between estimation accuracy, speed, and sequencing costs, taking into account the feasibility of inventorying a hyperdiverse group within a limited time frame (*Hebert et al., 2003*; *Dupuis et al., 2012*; *Eberle et al., 2020*). Several taxonomical works on Metriorrhynchini have simultaneously analysed mtDNA fragments, nuclear genes, morphology, and ecology (e.g. *Bocak and Yagi, 2010*; *Kalousova and Bocak, 2017*; *Bocek et al., 2019*; *Jiruskova et al., 2019*), but their results cannot robustly defend an application of a distance

threshold for the whole tribe. To avoid a possibility of diversity overestimation, we base further discussion on the 5% distance. We assume that such a threshold might be sufficiently cautious. Future taxonomic revisions are surely needed for the validation of here presented data on species diversity.

When analysing the *cox1* mtDNA fragment, we identified 1,848 mOTUs at 5% threshold and the numbers of delimited mOTUs indicate that, the substantial part of species diversity remains undescribed (*Table 2*, *Appendix 1—figure 14*). Additionally, the slope representing the relationship between the number of mOTUs and distance thresholds was gradual (*Appendix 1—figure 14*) due to a high number of genetically distant, indisputably distinct lineages in the dataset (*Appendix 1—figure 14*).

Our approach provides information about the diversity of the internal lineages. Metriorrhynchina is by far the most diverse group, within which the cladophorines comprise the largest clade (490 mOTUs, e.g, *Ditua* historically has included 2 spp., now ~250 spp.). The porrostomine clade is the next diverse group (373 mOTUs) and contains the speciose *Porrostoma* (126 mOTUs) and a paraphyletic series of lineages whose species have conventionally been placed in *Metriorrhynchus*. The differences between previously published data and our results are substantial and they question any reanalyses of literature data without prior verification (*Bocak, 2002*; *Bocak et al., 2020* references therein; *Source data 2 and 3*).

The numbers of mOTUs must be interpreted in the context of the sampling activity in each region. We did not use standardized protocols due to the long-term character of our research, the incorporation of some samples provided by other researchers, and the necessity to apply appropriate collecting methods to maximize the number of recorded species in various ecosystems and under different weather conditions. We identified only 94 mOTUs from the Afrotropical region, mainly due to the limited number of collecting trips by authors (64 localities) and the inaccessibility of some areas. Despite intensive field research (33 localities), we collected from the Philippines less than one third of the species described (33 mOTUs). Our collection activities in the Philippines were hindered by substantial loss of natural habitats, and this is soon expected to be the case in other regions (*Sodhi et al., 2004*). The number of species known from the Sundaland (114 localities) was approximately equal to the number of sequenced mOTUs despite disproportionately intensive collecting effort by the authors. Even after numerous expeditions to the Sundaland, many regions remain unsampled. As metriorrhynchine species ranges are small (*Jiruskova et al., 2019*; *Motyka et al., 2020*), the number of species will probably increase in the future. The proportion of new species was regionally ~70% if DNA data and morphology were considered in detailed taxonomic studies (e.g. *Jiruskova et al., 2019*). While these regions house numerous unknown species, we found New Guinea to be exceptionally diverse, with almost three times the number of species reported in the literature (1,105 mOTUs at 5% threshold; 175 localities, *Table 1* and *Table 2*). Despite the relatively large number of sampled localities, many areas of New Guinea remain unexplored and many places were only superficially sampled by colleagues and never visited by the authors (*Figure 1A*). Additional species were added to the dataset with each batch of sequenced samples from New Guinea and the area possibly houses much higher diversity than documented by the present study.

We observed a high turnover between regions, and few species had ranges which included landmasses separated by shallow seas (seven spp. Queensland / New Guinea, 9 spp. in Southeast Asia; *Source data 1*). Poorly dispersing lycids generally have very small ranges, except for the few genera that visit flowers and fly in open areas (*Motyka et al., 2021*). A similar small-scale turnover has recently been reported along altitudinal gradients (*Bocek et al., 2019*; *Motyka et al., 2020*; *Motyka et al., 2021*). A high turnover indicates a large proportion of hidden diversity, especially in tropical mountains (*Merckx et al., 2015*; *Mastretta-Yanes et al., 2018*). Mountain fauna is especially vulnerable to climate change and its inventorying is urgently needed.

The Metriorrhynchini has recently received considerable attention in taxonomic studies, and 302 species have been described by several authors over the past three decades, making a total of 1574 formally described species (*Table 1*, *Kazantsev, 2010*; *Bocak et al., 2020*; Appendix introductory information). Although the recent 24% increase in described diversity appears substantial, the distance-based analysis indicates the presence of almost 2000 mOTUs (*Appendix 1—figure 14*). An additional ~50 putative species (494 terminals) were identified, but this identification was only based on divergent morphology because of the absence of *cox1*. We assume that our sampling represents only a subset of all known species (< 50%). It means that the dataset contains ~1000 undescribed

species. At the current rate, formal morphological descriptions of an additional 1000 species would take decades or hundreds of years. This is a very long time in the context of the ongoing deforestation and fragmentation of natural habitats, and currently undocumented diversity might be lost long before it can be catalogued (*Brooks et al., 2002*; *Sodhi et al., 2004*; *Ceballos et al., 2015*; *Theng et al., 2020*). The rapid DNA-based inventory is an effective shortcut for obtaining basic information on the true diversity of tropical beetles and for setting a benchmark for future biodiversity re-evaluations.

The results reveal major biodiversity hotspots in New Guinea and the Sundaland. Tropical rainforests currently cover most of New Guinea, a tectonically young island that has not been considered a biodiversity hotspot for vertebrates (*Myers et al., 2000*; *Hall, 2011*; *Toussaint et al., 2014*). In the case of net-winged beetles, we show that the New Guinean fauna is phylogenetically diverse, spatially heterogeneous, and extremely rich as regards both the number of species and the endemic genera (*Table 1*). Additionally, the large clades of New Guinean species indicate that the diversification of major lineages preceded the uplift of the islands, and possibly started on the northern margin of the Australian craton and adjacent islands. Southeast Asia is a centre of phylogenetic diversity at the tribal level; its fauna contains all principal lineages and the highest diversity of Cautirina but is smaller than those of New Guinea. The Afrotropical and Palearctic regions represent only recently populated low-diversity outposts.

## Impact of biodiversity inventorying on biogeographical and evolutionary research

Detailed data on Metriorrhynchini diversity indicate low dispersal propensity and this makes Metriorrhynchini a promising model for biogeographic studies (*Ikeda et al., 2012*). Our densely sampled phylogeny did not find any long-distance dispersal events, in contrast to many studies of flying beetles (*Balke et al., 2009*; *Jordal, 2015*). Most recovered overseas dispersal events are limited to distances of less than 100 km and are commonly accompanied by speciation (*Source data 2 and 3*). The high

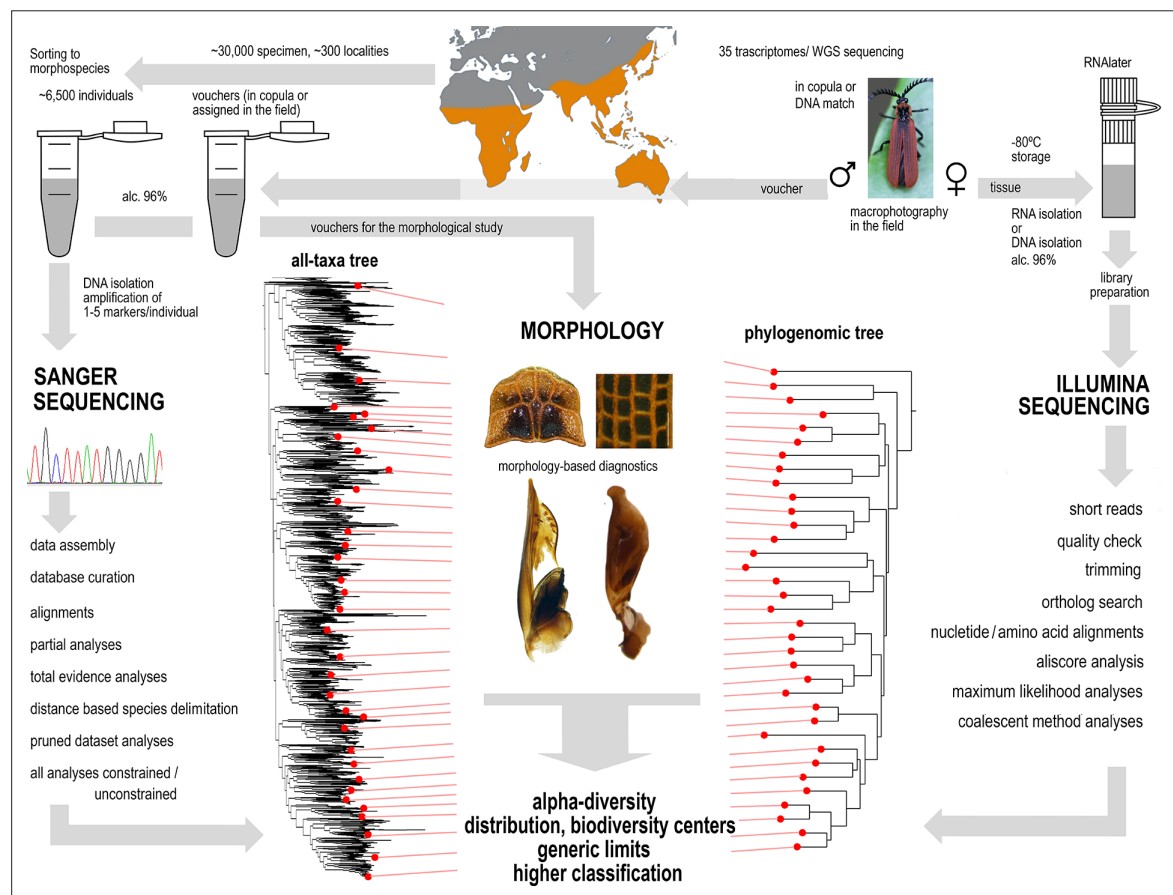

**Figure 5.** A sequence of applied methods from sampling to hypotheses.

proportion of erroneous placement of many taxa (*Appendix 1—figure 15*; *Bocak et al., 2020*) renders the distribution data cited in previous literature unsuitable for phylogeographic investigations, and revision of the classification is important in order to understand the true distribution of individual taxa. The original and revised ranges of selected genera are compared in *Appendix 1—figure 15* as examples.

Intensive biodiversity research has the potential to fill knowledge gaps concerning evolutionary phenomena that are mainly studied using a small number of model species, and the research can identify the unique attributes of other potential models. We document the contribution of a large-scale biodiversity inventory to evolutionary studies with two examples.

Net-winged beetles include several lineages in which females have lost the ability to completely metamorphose (*Bocak et al., 2008*; *Mcmahon and Hayward, 2016*). If a putative neotenic species is discovered, a comprehensive reference database of the group may identify its closest relatives. We used our data to place the East African *Cautires apterus* in a phylogenetic context, and the results indicated that it may be the youngest neotenic taxon of all net-winged beetles (36.1 my, *Figure 4*).

Our extensive DNA database of metriorrhynchine diversity may also play an important role in the study of the evolution of mimicry. Our inventory identified an extreme and previously unknown aposematic dimorphism in New Guinean metriorrhynchines (*Figure 4*; *Figure 5*). The placement of sexually dimorphic species in the phylogeny suggests that the shift to dimorphism was very recent (3.0 mya at the earliest) and began when both sexes were small-bodied. Mimetic sexual polymorphism is well understood in butterflies with non-mimetic males and mimetic females (*Kunte, 2008*), but the advergence of males and females to different aposematic models has only recently been reported in two subfamilies of net-winged beetles (*Motyka et al., 2018*; *Motyka et al., 2020*; *Motyka et al., 2021*). Divergent evolution in Müllerian systems appears to be more common in multi-pattern aposematic rings than was previously believed when morphology was the sole source of information.

## Conclusion

Priority areas for global conservation have usually been identified based on richness, species endemism and vulnerability of vertebrates (*Myers et al., 2000*; *Holt et al., 2013*). We assume that different patterns of biodiversity distribution can be revealed if other animal groups are studied. Reliable information on additional groups can focus our conservation efforts on valuable regions (*Morrison et al., 2009*; *Thomson et al., 2018*). Our model, beetles, is the most speciose group of animals but is much less known than vertebrates. Therefore, new data must be generated, and our research workflow must use innovations to economically produce the large-scale phylogenetic hypothesis for a high number of species. We conducted a worldwide sampling in ~700 localities, analysed transcriptomes, genomes, and mitochondrial markers, and validated our results with morphology. We show that the constrained position of less than 2% terminals increases the stability of tree topology and the congruence of molecular hypotheses with morphological traits (Appendix results). By the simultaneous consideration of genomic and mitochondrial phylogenetic signal, we achieved substantial progress with respect to the development of a Metriorrhynchini tree of life (*Chesters, 2017*; *Linard et al., 2016*). The voucher-based DNA entries established a framework for classifying samples from other studies, such as environmental sequencing (*Linard et al., 2016*; *Andújar et al., 2015*; *Arribas et al., 2016*) and for subsequent morphology-based studies. Despite limited time and funding, we identified almost 2000 mOTUs which indicate that there are at least twice more species than the number reported in the literature. This means that, at a conservative estimate, ~ 1000 species in the dataset were previously unknown to science. Furthermore, we identified New Guinea as a biodiversity hotspot, which is in clear contrast with studies identifying the biodiversity patterns of vertebrates. Our large-scale inventory shows that the literature records of tropical beetles cannot be used for biodiversity conservation and metanalyses without critical revision. We suggest that if focused field research is conducted even by a small research group and subsequent workflow steps are applied to any hyperdiverse tropical group, the results can set a benchmark for future evaluation of spatiotemporal changes in biodiversity.

## Materials and methods

### Field research

The analyzed individuals had been accumulated by numerous expeditions to various regions of the Metriorrhynchini range (*Figure 1A*, *Appendix 1—table 1*). The distribution of sampling sites was partly biased, and no samples are available from West Africa, Congo Basin, Sahel, Sri Lanka, and the Lesser Sundas. About 10% of samples were provided by other researchers.

Tissues for transcriptomic analyses were fixed in the field. As field identification is generally unreliable, we preferred to collect pairs *in copula*, then the female was fixed using RNAlater, and the male kept separately in 96% ethanol for Sanger sequencing and the voucher collection. Alternatively, the morphologically similar individual from the same place was fixed in ethanol and the identity of an individual assigned for transcriptomic analysis was confirmed by sequencing *cox1* mtDNA using tissue from the specimen preserved in RNAlater and putatively conspecific voucher (*Figure 2*). About 100 tissue samples were fixed and thirty-five of them were used for sequencing (*Appendix 1—table 2*). Earlier published transcriptomes were added (*McKenna et al., 2019*; *Kusy et al., 2019*). Due to the inaccessibility of properly fixed tissue, the two critical samples were shotgun sequenced using isolated DNA.

Almost 7000 samples from 696 localities (*Table 1*) were included in the sequencing program to obtain short mtDNA fragments. In total, 6429 yielding at least a single fragment were included in the analysis (*Supplementary file 1*). The analyzed data set contained some previously published sequences (e.g. *Sklenarova et al., 2013*; *Bocek and Bocak, 2019*). Voucher specimens are deposited in the collection of the Laboratory of Biodiversity and Molecular Evolution, CATRIN-CRH, Olomouc.

### Genomic and transcriptomic sequencing, data analysis

Libraries for thirty transcriptomes were prepared by Novogene Co., Ltd. (Beijing, China) and sequenced on the HiSeq X-ten platform (Illumina Inc, San Diego, CA). The removal of low-quality reads and TruSeq adaptor sequences were performed using fastp v.0.20.0 (*Chen et al., 2018*) with the following parameters: -q 5 u 50 l 50 n 15. All paired-end transcriptomic reads were assembled using SOAPdenovo-Trans-31mer (*Xie et al., 2014*).

Additionally, the total DNA (~33 Gb each) of *Metanoeus* sp. (Voucher code G19002) and an unidentified sample Metriorrhynchina species (Voucher JB0085) was shotgun-sequenced on the same platform. Reads were filtered with fastp using the same settings as above and quality was visualized with FastQC (http://www.bioinformatics.babraham.ac.uk/projects/fastqc). The draft genomes were assembled using SPAdes v.3.13.1 (*Bankevich et al., 2012*), with k-mer sizes of 21, 33, 55, 77, and 99. Obtained contigs were used to train Augustus (*Stanke and Waack, 2003*) for species-specific gene models with BUSCO (*Waterhouse et al., 2018*). Predicted species-specific gene models were then used for ab initio gene predictions in Augustus and predicted protein-coding sequences were used for subsequent analyses. Outgroup taxa were reported in previous studies (*Kusy et al., 2018*, *Kusy et al., 2019*; *McKenna et al., 2019*).

The ortholog set was collated by searching the OrthoDB 9.1 (*Zdobnov et al., 2017*) for single copy orthologs in six beetle genomes (*Appendix 1—table 3*). We used Orthograph v.0.6.3 (*Petersen et al., 2017*) with default settings to search in our assemblies for the presence of specified single copy orthologs. From the recovered 4193 orthologs, terminal stop codons were removed, and internal stop codons at the translational and nucleotide levels were masked. The amino acid sequences were aligned using MAFFT v.7.471 with the L-INS-i algorithm (*Katoh and Standley, 2013*). The alignments from each ortholog group were then checked for the presence of outliers. To identify random or ambiguous similarities within amino acid alignments, we used Aliscore v.2.076 with the maximum number of pairwise comparisons –r $10^{27}$, option -e. and we masked them using Alicut v.2.3 (*Kück et al., 2010*). Alinuc.pl was then used to apply the Aliscore results to match amino acids to the nucleotide data. MARE v.0.1.2-rc was used to calculate the information content of each gene partition (*Misof et al., 2013*). Partitions with zero information content were removed at both levels. Finally, the remaining 4109 alignments were retained for subsequent multispecies coalescent analyses, and different concatenated datasets were generated for both amino acid and nucleotide levels using FasConCat-G v.1.4 (*Kück and Longo, 2014*; *Appendix 1—table 4* and Appendix methods). The degree of missing data and overall completeness scores (Ca) across all datasets were inspected using AliStat v.1.7 (*Wong et al., 2020*).

## Compositional heterogeneity tests

To explore the effect of among species compositional heterogeneity and its possible bias to tree reconstruction, we inspected the data with BaCoCa v.1.105 (*Kück and Struck, 2014*) to identify the gene partitions that strongly deviate from compositional homogeneity using relative composition frequency variation value (RCFV). Following *Vasilikopoulos et al., 2019*, we considered compositional heterogeneity among species in a given partition to be high when RCFV ≥0.1. The heterogeneous partitions were excluded from the data to generate a more compositionally homogeneous dataset. We used Maximum Symmetry Test (*Naser-Khdour et al., 2019*) to identify the partitions that strongly deviate from compositional homogeneity at the nucleotide level (p-value cut off <0.05), and partitions below the threshold were excluded. The software SymTest v.2.0.49 (*Ott, 2019*) was used to calculate the overall deviation from stationarity, reversibility, and homogeneity (SRH) (*Ababneh et al., 2006*).

## Phylogenomic maximum likelihood analyses

For all datasets, phylogenetic reconstruction was performed using the maximum likelihood (ML) criterion with IQ-TREE 2.1.2 (*Minh et al., 2020*). First, we analyzed all datasets using the original gene partition boundary. The model selection for each gene was performed with ModelFinder (*Chernomor et al., 2016*; *Kalyaanamoorthy et al., 2017*) implemented in IQ-TREE2 (-MFP option). GTR model was considered for nucleotide supermatrices. For the amino acid supermatrices, the substitution models LG, DCMUT, JTT, JTTDCMUT, DAYHOFF, WAG, and free rate models LG4X and LG4M were tested. All possible combinations of modeling rate heterogeneity among sites were allowed (options: -mrate E,I,G,I + G,R -gmedian -merit BIC). We used the edge-linked partitioned model for tree reconstructions (-spp option) allowing each gene to have its own rate. The optimized partition schemes and best-fitting models were inferred for some datasets using -m MFP+ MERGE and the considering same substitution models as above. The fast-relaxed clustering algorithm was used to speed up computation during partition-scheme optimization (*Lanfear et al., 2017*). Ultrafast bootstrap (*Hoang et al., 2018*) and SH-like approximate likelihood ratio test (SH-aLRT) were calculated in IQ-TREE2 (options -bb 5000 and -alrt 5000) to assess nodal supports for focal relationships.

## Coalescent analyses and analyses of the confounding and alternative signal

To account for variation among gene trees owing to incomplete lineage sorting and to account for potential gene tree heterogeneity and discordance (*Edwards, 2009*), the data were also analyzed using the coalescent-based species-tree method. For every single gene partition, we calculated an ML gene tree in IQ-TREE2, with 5,000 ultrafast bootstrap replicates (-bb option) and using the same substitution models as predicted by ModelFinder in the above described partitioned analyses. For subsequent coalescent species tree estimation, the Accurate Species Tree Algorithm (ASTRAL-III v.5.7.3; *Zhang et al., 2018*) was used. To account for very poorly resolved branches on gene trees, branches with ultrafast bootstrap ≤10 were collapsed using newick utilities v.1.6 (*Junier and Zdobnov, 2010*) in every ASTRAL analysis. Local posterior probabilities (*Sayyari and Mirarab, 2016*) and quartet frequencies of the internal branches in every species tree were calculated using the parameter '-t = 2'. Pie charts representing quartet scores for the given topology and two alternatives were plotted to the resulting species trees in R using plot_Astral_trees (*Bellot, 2021*).

Additionally, we studied the effect of potentially confounding signals, like non-random distribution of data coverage and violations of SRH conditions, on our phylogenetic reconstructions with the Four-cluster likelihood mapping (FcLM) approach (*Strimmer and Haeseler, 1997*) implemented in IQ-TREE2. Based on the results of our tree reconstructions, we tested the hypotheses about the alternative placement of leptotrichaline and procautirine clades.

## Mitochondrial DNA sequencing and data analysis

Total DNA was extracted from the metathorax with a Wizard SV96 kit (Promega Corp., Madison, WI). The yield was measured using a NanoDrop-1000 Spectrophotometer (Thermo Fisher Scientific Inc, Waltham, MA). The PCR settings and cycle sequencing conditions were the same as those used by *Bocak et al., 2008*. Three fragments of mitochondrial genome were sequenced: *cox1*+ tRNA *Leu*+ *cox2* (~1100 bp), *rrnL*+ tRNA *Leu*+ *nad1* (~800 bp), and ~1210 bp of *nad5* and adjacent tRNA-*Phe*, tRNA-*Glu*, and tRNA-*Ser* mtDNA (the mtDNA fragments are further mentioned as *rrnL*, *cox1*, and

*nad5*). The PCR products were purified using PCRu96 Plates (Merck Millipore Inc, Burlington, MA) and sequenced by an ABI 3130 (Applied Biosystems, Waltham, MA) sequencer using the BigDye Terminator Cycle Sequencing Kit 1.1 (Applied Biosystems, Waltham, MA). Sequences were edited using Sequencher v.4.9 software (Gene Codes Corp., Ann Arbor, MI). Altogether 6476 individuals were analyzed including some previously published (*Sklenarova et al., 2013*; *Bocek and Bocak, 2019*).

The *cox1* gene fragment was used to OTUs delimitation (*Blaxter et al., 2005*) using CD-hit-est (*Fu et al., 2012*) and different thresholds (from similarity 0.99–0.90 by 0.05 steps). Therefore, we assembled two datasets: (A) the dataset containing all sequenced individuals and (B) all OTUs delineated by 0.98 similarity of the *cox1* gene. The *rrnL* and *tRNAs* were aligned using MAFFT 7.2 with Q-INS-I algorithm (*Katoh and Standley, 2013*), protein-coding genes were eye-checked for stop codons and aligned using Trans-Align (*Bininda-Emonds, 2005*). All fragments were concatenated using FasConCat (*Kück and Longo, 2014*) and analyzed under maximum-likelihood criterium in IQ-TREE v.2.1.2 (*Minh et al., 2020*; *Appendix 1—table 5*). To assess the branch supports values, we used SH-aLRT test with 1000 iterations. ModelFinder tool implemented in IQ-TREE was used to identify the best fit models using the Bayesian Information Criterion (*Chernomor et al., 2016*). The results of the TSA/WGS analyses were used to constrain basal topology among major clades of Metriorrhynchini in both analyses of datasets A and B. Further, we ran unconstrained analyses of the above-mentioned datasets with identical settings except -g option to compare results. We replicated constrained tree search nineteen-times and compared resulting trees using Robinson–Foulds distances in R package phangorn (*Schliep, 2011*; *Appendix 1—table 2*). Randomly chosen trees were then compared using cophylo script (phytools; *Revell, 2012*) with argument rotate = TRUE.

## Acknowledgements

Several colleagues provided samples from their respective home countries or their own field research. We are obliged to G Monteith and A Slipinski for their support during our research in Australia. The field research was enabled by the permits from the Government of Papua New Guinea, the Queensland Ministry of Environment, Malaysian Ministry of Natural Resources, and UP Los Baños. The trips to Papua New Guinea were supported by the Binatang Research Centre, Nagada and we are obliged to H Maraia and J Kua for their field assistance. A substantial part of the field research before the start of the Czech Science Foundation project was funded by the senior author and the Palacky University is acknowledged for granting the leave. L Harmackova advised with some analyses in R. We cordially acknowledge I Frebort and P Banas for their encouragement and support of this project after the liquidation of the laboratory at the Palacky University. Funding: This research was funded by The Czech Science Foundation, grant number 18–14942 S.

## Additional information

### Funding

| Funder | Grant reference number | Author |
| --- | --- | --- |
| Grantová Agentura České Republiky | 18-14942S | Ladislav Bocak |

The funders had no role in study design, data collection and interpretation, or the decision to submit the work for publication.

### Author contributions

Michal Motyka, Conceptualization, Data curation, Formal analysis, Investigation, Methodology, Resources, Visualization, Writing – original draft, Writing – review and editing; Dominik Kusy, Conceptualization, Data curation, Formal analysis, Methodology, Resources, Visualization, Writing – original draft, Writing – review and editing; Matej Bocek, Renata Bilkova, Data curation, Investigation, Resources, Writing – review and editing; Ladislav Bocak, Conceptualization, Data curation, Funding acquisition, Project administration, Resources, Supervision, Writing – original draft, Writing – review and editing

**Author ORCIDs**
Matej Bocek (iD) https://orcid.org/0000-0002-3398-6078
Ladislav Bocak (iD) https://orcid.org/0000-0001-6382-8006

**Decision letter and Author response**
Decision letter https://doi.org/10.7554/eLife.71895.sa1
Author response https://doi.org/10.7554/eLife.71895.sa2

## Additional files

### Supplementary files
Transparent reporting form

Supplementary file 1. Maximum likelihood tree recovered by the analysis of the full dataset (mitochondrial fragments); numbers above branches represent SH-alrt support values.

Source data 1. Maximum likelihood tree recovered by the analysis of the full dataset (mitochondrial fragments). Depicted numbers above branches represent SH-alrt support values.

Source data 2. Maximum likelihood tree recovered by the analysis of the reduced dataset (98% similarity OTUs). Depicted numbers above branches represent SH-aLRT support values. The taxa with the constrained position in the tree are marked with the red star.

Source data 3. Maximum likelihood tree recovered by the analysis of the reduced dataset (mitochondrial fragments, 98% similarity OTUs) with unconstrained backbone. Depicted numbers above branches represent SH-alrt support values.

### Data availability
All datasets are deposited in the Mendeley Data repository https://doi.org/10.17632/ntgg6k4fjx.1.

The following dataset was generated:

| Author(s) | Year | Dataset title | Dataset URL | Database and Identifier |
|---|---|---|---|---|
| Motyka M, Kusy D, Bocek M, Bilkova R, Bocak L | 2021 | Data for: Phylogenomic and mitogenomic data can accelerate inventorying of tropical beetles during the current biodiversity crisis | https://doi.org/10.17632/ntgg6k4fjx.1 | Mendeley Data, 10.17632/ntgg6k4fjx.1 |

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

# Appendix 1

## Appendix introductory information

### Overview of Metriorrhynchini systematics and natural history

Lycidae (net-winged beetles) is a diverse, mostly tropical family of Elateroidea and Metriorrhynchini is the largest lycid tribe with almost 1600 species (*Masek et al., 2018*). The history of their classification started with Kleine's proposals of tribes and/or subfamilies Metriorrhynchini/inae, Trichalini/inae, Cladophorini/inae, and Dilolycini/inae (see an overview by *Kleine, 1933*). The supergeneric classification of net-winged beetles was later revised. Conderini were merged with Metriorrhynchini and Trichalini in the widely defined Metriorrhynchinae and Cladophorini and Dilolycini were synonymized to Metriorrhynchini (*Bocak, 2002* and references therein). *Calder, 1998* considered Metriorhynchinae Kleine, 1926 (based on *Metriorhynchus* Guérin-Méneville, 1838) to be a homonym of the crocodilian subfamily Metriorhynchinae Meyer, 1832 and used Dilolycinae Kleine, 1926 as a replacement name. Trichalinae were placed as a tribe Trichalini in Metriorrhynchinae or a subtribe Trichalina in Metriorrhynchini (*Bocak, 2002*). Additionally, Hemiconderina *Bocak et al., 2008* were erected for *Hemiconderis* and related genera. Detailed information on the metriorrhynchine classification history was published in the latest revision of the subtribal classification (*Sklenarova et al., 2014*) and the genera were assigned to subtribes by *Kubecek et al., 2015* and *Bocak et al., 2020*.

The alpha-taxonomy is mostly based on the original, often uninformative descriptions (catalogued by *Kleine, 1933*; *Bocak et al., 2020*). Recently, *Calder, 1998* compiled the catalogue of Australian net-winged beetles and proposed *Metriorhynchus* Guérin-Méneville, 1838 as a junior homonym of *Metriorhynchus* Meyer, 1832. Subsequently, *Porrostoma* Castelnau, 1838 was used as a replacement name for *Metriorhynchus* Guérin-Méneville, 1838 and new combinations and new replacement names were proposed (*Calder, 1998*). *Bocak, 2002* reviewed the status of lycid taxa described by *Guérin-Méneville, 1838* and proposed to replace *Metriorhynchus* Guérin-Méneville, 1838 by *Metriorrhynchus* Gemminger et Harold, 1869. *Bocak, 2002* additionally considered *Porrostoma* Castelnau, 1838 as a valid name for a separate genus in Metriorrhynchini. These studies resulted in repeated transfers of several hundred of species between *Metriorrhynchus* and *Porrostoma* (*Calder, 1998*; *Bocak et al., 2020*).

There are five Metriorrhynchini genera which house the majority of described species: *Xylobanus* (231 spp.), *Cautires* (437 spp.), *Trichalus* (120 spp.) *Metriorrhynchus* (194 spp.), and *Cladophorus* (131 spp.; *Kleine, 1933*; *Calder, 1998*; *Bocak, 2002*; *Bocak et al., 2020*), although their limits have been ambiguous. *Trichalus* was recovered as a much smaller clade than *Microtrichalus* Pic, 1921 (*Bocek and Bocak, 2017*). Australian *Metriorrhynchus* transferred to *Porrostoma* Castelnau, 1838 by *Calder, 1998* were formally returned to Metriorrhynchus by *Metriorrhynchus* by *Bocak et al., 2020*. *Xylobanus* was recovered as a clade that contains most Oriental Cautirina with four primary costae, but Afrotropical *Xylobanus* were recovered as a distantly related terminal clade within Afrotropical *Cautires* (*Sklenarova et al., 2013*; *Sklenarova et al., 2014*). Some Oriental *Xylobanus* belong to Metanoeina and New Guinean species to Metriorrhynchina (*Kubecek et al., 2015*; *Bocak et al., 2020*). An additional typologically defined genus, *Procautires* Kleine, 1925, was reported from Asia, Africa and Australasia (*Kleine, 1933*), but the type species belongs to Metriorrhynchina in contrast to the Afrotropical and Asian species which are close to *Cautires* (Cautirina).

### Geographic distribution

The tribe Metriorrhynchini dominates the Australian net-winged beetle fauna with ~85% (196 spp.) of the named taxa. They are similarly common in New Guinea (423 spp.) and the Wallacea (176 spp.). Although highly diversified (261 spp. and 231 spp., respectively), they represent only about 30% of described net-winged beetle species in the Oriental and Afrotropical regions where similar numbers of species belong to Platerodini, Calochromini and Lycini, or several small tribes (*Masek et al., 2018*).

The tribe Metriorrhynchini has a Gondwanan distribution, and the range reaches to the Palearctic East Asia (Russian Far East, Cautirina and Metanoeina). Additionally, the tribe is well represented in the Sino-Japanese realm sensu *Holt et al., 2013* (China, Japan, Cautirina and Metanoeina in the whole area, Metriorrhynchina in southernmost China only) and in the Oriental realm (most of India, Sri Lanka, the lower elevations of the Himalayas, Indo-Burma, Malaya, the Great and Lesser Sundas,

the Philippines). Further, the tribe is widespread in the Oceanian realm, especially in its western part (the Moluccas, New Guinea, the Solomon Islands and a few islands further east) and in the Australian realm (Australia, Tasmania, and one introduced species in New Zealand). The Afrotropical realm houses relatively diversified fauna in forest and savannah habitats, but all species belong to Cautirina and represent only two clades which correspond with putative independent colonization events from drifting India to Madagascar and continental Africa (*Sklenarova et al., 2013*).

The subtribes differ in their distribution. Metriorrhynchina is mostly Oceanian and Australian, relatively rich Metriorrhynchina fauna is known from the Sulawesi, much less species occur in the Philippines and only a few species of *Diatrichalus*, *Leptotrichalus*, *Trichalus*, *Microtrichalus*, and *Metriorrhynchus* in the eastern part of the Oriental realm (*Bocak and Yagi, 2010*; *Bocek and Bocak, 2019*; *Bocek et al., 2019*) and southern Yunnan in the Sino-Japanese realm. The Cautirina occurs in the Afrotropical, Oriental, Palearctic, and Sino-Japanese realms, but they do not cross Weber's line that was proposed as a border between Oriental and Oceanian realm (*Holt et al., 2013*). The Metanoeina is Oriental, with a limited number of species in China and Japan. Only *Metanoeus* occurs in the Philippines.

The Metriorrhynchini is the species-rich tribe and the almost 200 years of taxonomic research accumulated data that need serious revision (*Bocak et al., 2020*). In the taxonomy of this group, we face the burden of uninformative descriptions (e.g. the studies by the French entomologist M. Pic in the first half of the 20th century). The careless classification of newly described species, inaccessibility of type-holding collections or their poor organization, and legal regulations limit taxonomic research (*Prathapan, 2018*; *Bocak et al., 2020*).

## Biology

The adults are volatile, but they are poor dispersers due to their soft-bodiedness and many species remain in shaded situations under the canopy, usually sitting on leaves. Only some species (*Porrostoma*, *Metriorrhynchus*, *Leptotrichalus*, and *Trichalus*) visit flowers. The flower frequenting species are more common in semi-dry areas where nectar represents a source of water for adults instead of water on leaves or in rotten wood. All Metriorrhynchini are unpalatable for predators and most are aposematically coloured (*Eisner et al., 2008*; *Motyka et al., 2021*). The color patterns are usually geographically restricted and the sympatrically occurring species form Müllerian mimicry rings (*Bocak and Yagi, 2010*; *Motyka et al., 2018*; *Motyka et al., 2020*; *Motyka et al., 2021*; *Bocek et al., 2019*). The larvae of *Metanoeus*, *Cautires*, *Leptotrichalus*, *Metriorrhynchus*, and *Porrostoma* were described by *Bocak and Matsuda, 2003*. They live in rotten wood including trunks and logs on soil surface or in rotten roots in soil in arid regions.

## Appendix results

Metriorrhynchina subclades based on phylogenomic analyses and their morphological characteristics and distribution.

### The procautirine clade

The phylogenomic analysis contained four taxa that clustered as the first or second deepest serial branch in Metriorrhynchina. One of the internal clades contains species that are morphologically highly similar to the type-species of *Procautires*. They have reduced secondary elytral costae in the middle part of the elytra. Other procautirine species have fully developed secondary costae unlike *Procautires*. The only morphological trait supporting the monophyly of the procautirine clade is a setose patch in the internal sac of the male genitalia.

### The leptotrichaline clade

The phylogenomic backbone identified close relationships of three endemic genera from the Sulawesi (*Mangkutanus*, *Wakarumbia*, and *Broxylus*), and two genera, *Sulabanus*, *Leptotrichalus*, known from Sulawesi and the Philippines (*Bocak et al., 2020*). The mtDNA dataset identified as closely related the Australian and New Guinean *Synchonnus* and New Guinean *Falsolucidota*. Two lineages of the clade putatively colonized the Oriental region: *Leptotrichalus* (the Philippines, Sundas, Indo-Burma and southernmost China) and *Sulabanus* (only the Philippines, 2 spp. from Mindanao and Sibuyan identified herein, either undescribed species or currently placed in Cautirina: *Xylobanus*). We have not found any reliable external diagnostic morphological character which would support the relationships of these genera. Only *Sulabanus* and *Mangkutanus* have a complete set of lateral

pronotal carinae. Other genera have simplified of absent carinae except those that form the median lanceolate areola (**Bocak, 2002**). The group contains genera with the highly diverse arrangement of elytral costae.

## The trichaline clade

The genomic dataset contained representatives of three genera: *Diatrichalus*, *Microtrichalus*, and *Eniclases*, all of them sharing the characteristic morphological traits of the trichaline clade, that is a single median areola in the pronotum and the shortened first primary elytral costa. We identified as close relatives two lineages of small-bodied Metriorrhynchina with conspicuous median pronotal areola and either absent or simplified lateral carinae, but with the full-length elytral costa 1. The structure of elytral costae is diverse in the clade: most trichalines sensu stricto have well-developed secondary costae, that is, nine costae at humeri, only a few *Diatrichalus* have only four primary costae in the humeral part. All genera of the here delimited trichaline clade, except *Diatrichalus*, share similar male genitalia (**Bocak, 2002**). The deepest branches of the trichaline clade contain mostly New Guinean species and several lineages of the trichalines colonized the Sulawesi, Philippines, Greater Sundas, Malaya, Indo-Burma, and southernmost China.

## The porrostomine clade

Six transcriptomes were analyzed to recover the backbone of the clade. *Porrostoma* and *Metriorrhynchus* are distantly related and easy to distinguish with the genital morphology (**Bocak, 2002**). Numerous small bodied, morphologically diverse Metriorrhynchina species were identified as their close relatives. Most *Metriorrhynchus* and their closest relatives come from New Guinea, but several species of a terminal clade crossed Lydderker's, Weber's, and Wallace's lines and colonized Sulawesi, the Philippines, Greater Sundas, Malaya, and Indo-Burma (**Bocak and Yagi, 2010**).

## The cladophorine clade

Altogether eleven taxa were included in the genomic analysis, and we found *Cladophorus* as a sister to *Pseudodontocerus* (=*Carathrix*) and *Ditua*. Most genera have seven pronotal areoles, flabellate to pectinate male antennae, and nine well developed costae in the elytra. Mitochondrial dataset recovered this clade as the most speciose group. The characteristic large-bodied species resembling the type-species of *Cladophorus* were found as a terminal subclade among several lineages of small-bodied Metriorrhynchina which do not resemble them in general appearance. Further splits contain the *Pseudodontocerus* clade and a large clade containing representatives of earlier described genera *Ditua* and *Cautiromimus*. The species of the *Cladophorus* clade are most diverse in New Guinea, but some species of the *Ditua* clade were recorded from the Philippines and Sulawesi, a limited number of species is native to continental Australia.

## Impact of molecular phylogenetics on the assessment of morphological evolution

With the robust, data-rich phylogeny, we can test earlier hypotheses on morphological evolution. Relationships among subtribes have been based on the similarity of larvae and ambiguously supported by Sanger data (**Sklenarova et al., 2013**; **Sklenarova et al., 2014**). Our results imply that bipartite larval terga can be inferred either as a synapomorphy of Metriorrhynchini with reversal to entire tergites in Cautirina or we must alternatively consider their independent origin in Metanoeina and Metriorrhynchina. The mapping of the character on the phylogenomic tree needs two instead of a single step as earlier supposed (**Sklenarova et al., 2014**). When most larvae are unknown and their morphology variable, the phylogenetic hypotheses based on incomplete morphological data and uncertain polarity urgently need validation from independent sources of information.

Due to quite high morphological disparity and ambiguous interpretation of some morphological characters, there were proposed subfamilies, tribes, and/or subtribes for various subsets of genera now placed in the Metriorrhynchina (Metriorrhynchinae/ini, Cladophorinae/ini, Trichalinae/ini, Dilolycinae/ini, Hemiconderina, and possibly Melanerotini, now in Lycidae *incertae sedis*; **Bocak, 2002**; **Kazantsev, 2010**; **Kusy et al., 2019**). Phylogenomics reject the possibility to assign a high rank to Trichalini and Hemiconderina as numerous coordinated taxa would have to be erected to make all groups reciprocally monophyletic and keep the earlier proposed names valid.

The phylogenomic analyses reinforce the previously identified monophyly of the trichaline clade (=earlier Trichalini part; **Bocek and Bocak, 2017**), but simultaneously suggest their delayed origin

as a terminal clade and the presence of some 'non-trichaline' lineages as serial sister-groups to the trichalines in the earlier sense. The cladophorine clade contains numerous taxa which had been placed until recently into *Cautires* (Cautirina), a distant lineage in respect to cladophorines. Herein, we further delimit the porrostomine clade which contains *Metriorrhynchus* and *Porrostoma*, but along with them several deeply rooted lineages which have been incorrectly placed to various genera.

For the first time, we define the leptotrichaline clade that contains the genera earlier placed into the trichalines (*Leptotrichalus*), Hemiconderina (*Falsolucidota*), Lycinae: Calopterini (*Broxylus*), and Cautirina (some *Xylobanus*). The procautirine clade is also a newly defined group. *Procautires* had been for long considered as related to *Cautires*, but it is one of earliest Metriorrhynchina.

Until recently, the classification had been based almost exclusively on the arrangement of pronotal carinae, elytral costae and the shape of antennae. These traits are easily observable and sometimes define a clade, but in many cases their presence and/or absence is misleading as they are highly plastic even within restricted clusters of species (*Sklenarova et al., 2014*; *Bocek and Bocak, 2017*; *Kusy et al., 2019*). Before the availability of molecular data, the homoplasy of external morphological characters could not be tested. As a result, the earlier studies coped with uncertain relationships of newly described species at the levels from a tribe down to a genus (*Kleine, 1933*; *Calder, 1998*; *Bocak et al., 2020*).

The list of Metriorrhynchini genera with complete synonymy:

Subtribe Metanoeina Sklenarova et al., 2014
*Matsudanoeus* Sklenarova et al., 2014
*Metanoeus* Waterhouse, 1878
*Ochinoeus* Kubecek et al., 2015
*Xylometanoeus* Sklenarova et al. 2014
Subtribe Cautirina Sklenarova et al. 2014
Subtribe Cautirina Sklenarova et al. 2015
*Caenioxylobanus* Waterhouse, 1878
*Cautires* Waterhouse, 1878
*Prometanoeus* Kleine, 1925
= *Tapromenoeus* Bocak et Bocakova, 1989
*Paracautires* Kazantsev, 2012
*Spartoires* Kazantsev, 2012
*Tricautires* Kazantsev, 2006
*Xylobanus* Waterhouse, 1878
Subtribe Metriorrhynchina Kleine, 1926
Procautirine clade
*Procautires* Kleine, 1925b
*Xylothrix* Kazantsev, 2015
Leptotrichaline clade
*Falsolucidota* Pic, 1921
= *Hemiconderis* Kleine, 1926
*Broxylus* Waterhouse, 1879
= *Samanga* Pic, 1921
*Mangkutanus* Kubecek et al., 2011
*Sulabanus* Dvorak & Bocak, 2007
*Leptotrichalus* Kleine, 1925
*Synchonnus* Waterhouse, 1879
= *Achras* Waterhouse, 1879
= *Enylus* Waterhouse, 1879
= *Strophicus* Waterhouse, 1879
*Wakarumbia* Bocak, 1999
Cladophorine clade
*Cautiromimus* Pic, 1926
*Cladophorinus* Kleine, 1926
*Cladophorus* Guérin-Méneville, 1830

= *Odontocerus* Guérin-Méneville, 1838
= *Spacekia* Strand, 1936
subgenus *Cladophorus* s. str.
subgenus *Falsocautires* Pic, 1926
*Ditua* Waterhouse, 1879
*Marena* Kazantsev, 2007
*Pseudodontocerus* Pic, 1921
= *Carathrix* Kleine, 1926
Trichaline clade
*Diatrichalus* Kleine, 1926
= *Mimoxylobanus* Pic, 1921
*Eniclases* Waterhouse, 1879
= *Trichalolus* Pic, 1923
*Flabellotrichalus* Pic, 1921
= *Stereotrichalus* Kleine, 1926
= *Villosotrichalus* Pic, 1921
subgen. *Flabellotrichalus* s. str.
subgen. *Maibrius* Bocek et Bocak, 2017
*Lobatang* Bocak, 1998
subgenus *Lobatang* s. str.
subgenus *Spinotrichalus* Kazantsev, 2010
*Microtrichalus* Pic, 1921
= *Falsoenylus* Pic, 1926
*Schizotrichalus* Kleine, 1926
*Trichalus* Waterhouse, 1877
= *Xantheros* Fairmaire, 1877
Porrostomine clade
*Metriorrhynchus* Gemminger et Harold, 1869
= *Metriorhynchus* Guérin-Méneville, 1838
= *Dilolycus* Kleine, 1926
= *Flabelloporrostoma* Pic, 1923
*Porrostoma* Laporte, 1838
*Oriomum* Bocak, 1999
*Metriorrhynchoides* Kleine, 1926
*Stadenus* Waterhouse, 1879
Metriorrhynchina *incertae sedis*
*Malacolycus* Kleine, 1943
*Xylobanomimus* Kleine, 1926
*Xylobanomorphus* Kleine, 1935
*Kassemia* Bocak, 1998

## Appendix methods

### Genomic and transcriptomic sequencing, data analysis, individual datasets

Libraries for thirty transcriptomes were prepared by Novogene Co., Ltd. (Beijing, China) and sequenced on the HiSeq X-ten platform (Illumina Inc, San Diego, USA). The removal of low-quality reads and TruSeq adaptor sequences were performed using fastp v.0.20.0 (*Chen et al., 2018*) with the following parameters: -q 5 u 50 l 50 n 15. All paired-end transcriptomic reads were assembled using SOAPdenovo-Trans-31mer (*Xie et al., 2014*).

Additionally, the total DNA (~33 Gb each) of *Metanoeus* sp. and an unidentified sample JB0085 was shotgun-sequenced on the same platform by Novogene Co., Ltd. (Beijing, China) for 150 bp paired-end reads. Read were filtered with fastp using same setting as above and quality was visualized with FastQC (http://www.bioinformatics.babraham.ac.uk/projects/fastqc).

The draft genomes were assembled using SPAdes v.3.13.1 (*Bankevich et al., 2012*), with all parameters set to default values and k-mer sizes of 21, 33, 55, 77 and 99. Obtained contig sequences were used to train Augustus (*Stanke and Waack, 2003*) for species specific gene models

with BUSCO v.3 (Waterhouse et al., 2018), -long option, Endopterygota set of conserved genes (n = 2442) and -sp tribolium2012 as the closest relative. Predicted species specific gene models were then used for *ab initio* gene predictions in Augustus and predicted protein coding sequences were used for subsequent analyses. Seven outgroup taxa, *Porrostoma* and *Mangkutanus* transcriptomes were reported in previous studies (*Kusy et al., 2018*, *Kusy et al., 2019*; *McKenna et al., 2019*). The filtering and assembling methods for these samples have been described by *Kusy et al., 2019*.

The ortholog set was collated by searching the OrthoDB v.9.1 database (*Zdobnov et al., 2017*) for single copy orthologs in six beetle genomes *Onthophagus taurus* (*Poelchau et al., 2015*), *Tribolium castaneum* (*Shelton et al., 2015*; *Richards et al., 2008*) *Dendroctonus ponderosae* (*Keeling et al., 2013*), *Anoplophora glabripennis* (*McKenna et al., 2017*), *Leptinotarsa decemlineata* (*Poelchau et al., 2015*) and *Agrilus planipennis* (*Poelchau et al., 2015*; *Appendix 1—table 3*). OrthoDB v.9.1 predicted 4225 single copy orthologs for beetle species and Coleoptera (Polyphaga) reference node. We used Orthograph v.0.6.3 (*Petersen et al., 2017*) with default settings to search in our assemblies for the presence of specified single copy orthologs. From the recovered 4193 orthologs, terminal stop codons were removed, and internal stop codons at the translational and nucleotide levels were masked using the Perl script summarize_orthograph_results.pl (*Petersen et al., 2017*). The amino acid sequences were aligned using MAFFT v.7.471 with the L-INS-i algorithm (*Katoh and Standley, 2013*). The alignments from each ortholog group were then checked for the presence of outliers using the script checker_complete.1.3.1.2.pl, according to previously published methods (*Misof et al., 2014*; *Peters et al., 2017*). Identified outlier sequences were removed from amino acid and nucleotide alignments. After this step, the sequences of reference taxa and all gap-only sites were pruned. Corresponding multiple sequence alignments of nucleotides were generated using Pal2Nal (*Suyama et al., 2006*). To identify random or ambiguous similarities within amino acid alignments, we used Aliscore v. 2.076 with the maximum number of pairwise comparisons –r 1027, a special scoring approach for gap-filled amino acid sites option -e, and other parameters set to default values. Any random or ambiguous similarities were masked using Alicut 2.3 (*Kück et al., 2010*). Alinuc.pl was then used to apply the Aliscore results to match amino acids to the nucleotide data (*Peters et al., 2017*). Additionally, in each gene alignment, the short randomly aligned fragments were replaced with gaps and sequences with ≥80% missing data (calculated as percentage of '-' and 'X' in the amino acid alignments and '-' and 'N' in the nucleotide alignments) were removed using Python scripts (*Zhang, 2021*). MARE v. 0.1.2-rc was used to calculate the information content of each gene partition in terms of amino acid coding (*Misof et al., 2013*). Partitions with zero information content (IC0) were removed at both amino acid and nucleotide level.

Finally, the remaining 4,109 alignments were retained for subsequent multispecies coalescent analyses, and different concatenated datasets were generated for both amino acid and nucleotide levels using FasConCat-G v.1.4 (*Kück and Longo, 2014*). With all 4109 genes, we generated datasets designated as A-4199-AA and B-4199- NT (name-number of partitions-level). To reduce the effect of saturation on phylogenetic reconstruction (*Breinholt and Kawahara, 2013*) we created the dataset C-4199-NT12 and E-4109-NT2, with third-codon or third and first-codon positions excluded. The degree of missing data and overall completeness scores (Ca) across all datasets were inspected using AliStat v.1.7 (*Wong, 2021*) and heatmaps of pairwise completeness scores for all analysed datasets were generated (see *Appendix 1—table 3*). Further Information content and saturation values (the overall degree of data coverage with respect to gene presence or absence) were calculated with MARE v.0.1.2-rc (MAtrix REduction) (*Misof et al., 2013*) for each dataset.

## Compositional heterogeneity tests

To explore the effect of among species compositional heterogeneity and its possible bias to tree reconstruction, we inspected the data with BaCoCa v.1.105 (*Kück and Struck, 2014*) to identify the gene partitions that strongly deviate from compositional homogeneity using relative composition frequency variation value (RCFV). Following *Fernández et al., 2016* and *Vasilikopoulos et al., 2019*, we considered compositional heterogeneity among species in a partition to be high when the overall RCFV value was ≥0.1. The compositionally heterogeneous partitions were excluded from the dataset A-4109-AA to generate more compositionally homogeneous dataset D- 3370-AA. To create more homogenous and decisive dataset, we reduced the dataset D-3370-AA to contain only partitions with all taxa and created dataset F-1490-AA. Additionally, we used MARE to increase information content of dataset D-3370-AA and default output supermatrix is designed as dataset

J-2129-AA_MARE. We used tree matched-pairs tests of homogeneity the MPTS (matched- pairs test of symmetry), MPTMS (matched-pairs test of marginal symmetry), and MPTIS (matched- pairs test of internal symmetry) (*Naser-Khdour et al., 2019*) implemented in IQ-TREE2 to identify the partitions that strongly deviate from compositional homogeneity at nucleotide level (P-value cutoff <0.05) and partitions below the threshold were excluded. And following datasets generated: G-NT-1767_ MaxSymTest, H-NT-1645_MaxSymTestmarginal and I-NT-3905_MaxSymTestInternal.

The software SymTest v.2.0.49 (https://github.com/ottmi/symtest) was used to calculate the overall deviation from stationarity, reversibility, and homogeneity (SRH) (*Ababneh et al., 2006*) between the amino acid and nucleotide sequences. Heatmaps were generated for all datasets to visualize the pairwise deviations from SRH conditions.

## Phylogenetic maximum likelihood analyses

For all datasets phylogenetic reconstruction was performed using maximum likelihood (ML) criterion with IQ-TREE 2.1.2 (*Minh et al., 2020*). First, we analyzed all datasets using original gene partition boundary. The model selection for each gene was performed with ModelFinder (*Chernomor et al., 2016*; *Kalyaanamoorthy et al., 2017*) implemented in IQ-TREE2 (-MFP option). GTR model were considered for nucleotide supermatrices. For the amino acid supermatrices, the substitution models LG, DCMUT, JTT, JTTDCMUT, DAYHOFF, WAG and free rate models LG4X and LG4M were tested. All possible combinations of modeling rate heterogeneity among sites were allowed (options: -mrate E,I,G,I + G,R -gmedian -merit BIC). We used the edge-linked partitioned model for tree reconstructions (-spp option) allowing each gene to have its own rate. The optimized partition schemes and best-fitting models were inferred for dataset A-4109-AA using -m MFP+ MERGE and considering same substitution models as above. The fast-relaxed clustering algorithm was used to speed up computation during partition-scheme optimization (*Lanfear et al., 2017*). The top 10% of partitions schemes `--rclusterf 10` and maximum 12 327 partitions pairs (three times the number of partitions) `--rcluster-max 12 327` were considered. Dataset A-4109-AA was also analyzed without any prior partitioning and under best fitting JTT + F + R9 substitution model. Ultrafast bootstrap (*Hoang et al., 2018*) and SH-like approximate likelihood ratio test (SH-aLRT) were calculated in IQ-TREE2 (options -bb 5000 and -alrt 5000) to assess nodal supports for focal relationships (*Appendix 1—table 3*).

Analyses of the confounding and alternative signal with four-cluster likelihood mapping (FcLM) and sequence data permutations

Additionally, we studied the effect of potentially confounding signal, like non-random distribution of data coverage and violations of SRH conditions, on our phylogenetic reconstructions with the Four- cluster likelihood mapping (FcLM) approach (*Strimmer and Haeseler, 1997*) as described by *Misof et al., 2014* and implemented in IQ-TREE2. Based on the results of our ML tree reconstructions we tested the hypotheses about alternative placement of the leptotrichaline and procautirine clades. Permutation of data were executed as described in *Misof et al., 2014* Our approach only differs in the use of the LG substitution matrix (*Le and Gascuel, 2008*) for permuting the supermatrices instead of WAG substitution matrix. For the analysis of the permuted supermatrices we used the option -q for the partition model in IQ-TREE2 and we used the LG model (for amino-acid alignments) and GTR model (for the nucleotide alignments). With original data, we performed the FcLM analyses by applying previously estimated substitution models (edgelinked models, -spp) with IQ-TREE2.

For estimation of *Cautires apterus* and sexually dimorphic *Metriorrhynchus* sp. splits were used subsets of the samples which represent the most common lineages. Dating of forementioned diversification was analyzed in BEAST v.1.8.1 (*Drummond et al., 2012*) applying HKY + I + G substitution model, uncorrelated relaxed clock model and birth–death process tree prior and 0.0115 rate for *cox1* gene as applied earlier by *Bocak and Yagi, 2010*. The stationary phase was checked in Tracer v.1.5 (http://beast.bio.ed.ac.uk/Tracer). The consensus tree was computed with TreeAnnotator v.1.8.2 (http://beast.bio.ed.ac.uk/treeannotator) after eliminating a part of trees as a burn-in after evaluating ESS in tracer v.1.5. The final trees were visualised in Figtree v.1.4.4.

Data deposition: The additional supporting data and all the analysed supermatrices are deposited in the Mendeley Data repository DOI:https://doi.org/10.17632/ntgg6k4fjx.1.

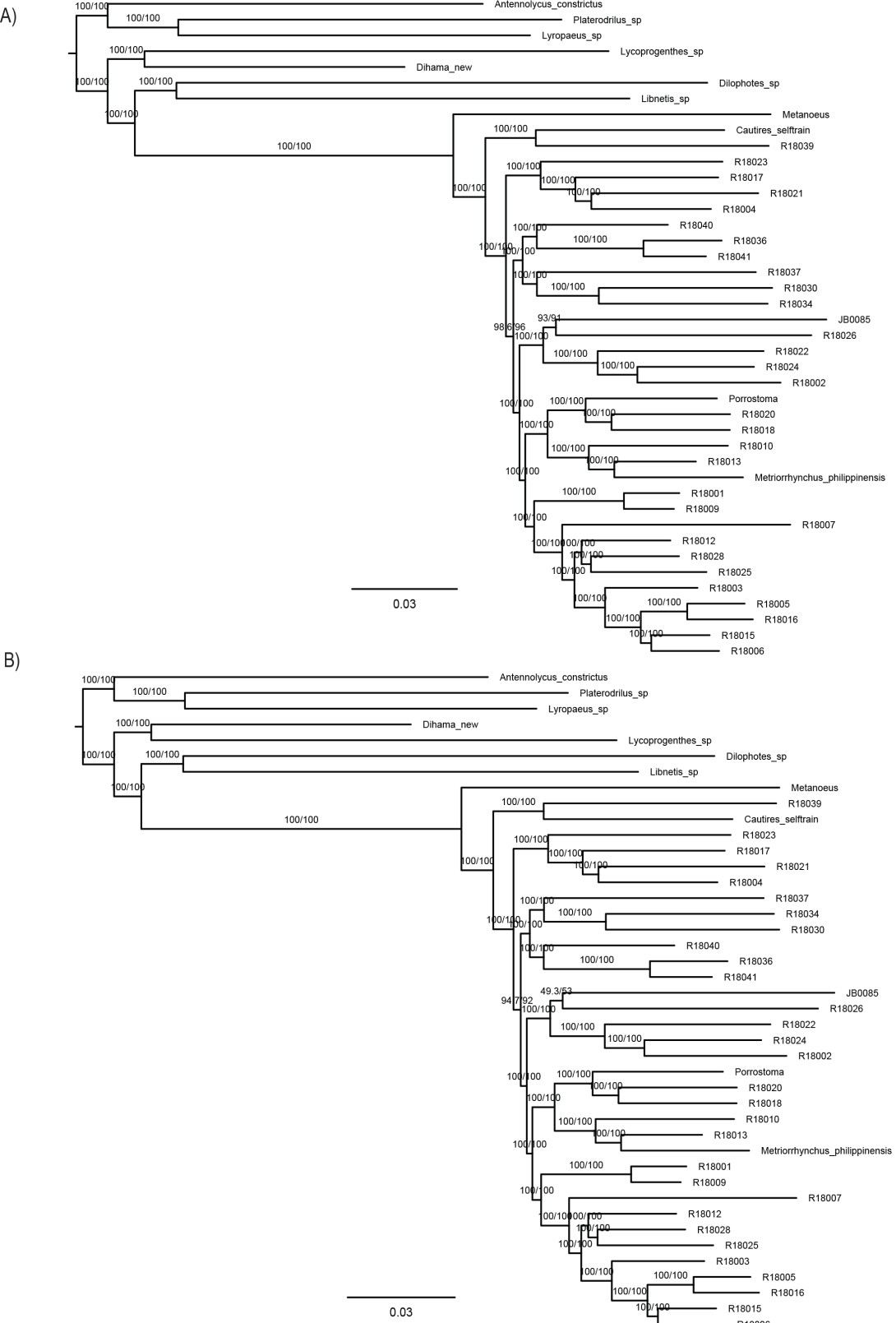

**Appendix 1—figure 1.** Maximum likelihood trees from IQ-TREE amino acid analysis of dataset A –4109-AA. (**A**) With partitioning by gene and (**B**) without partitioning. The depicted branch support values represent SH-aLRT and ultrafast bootstrap.

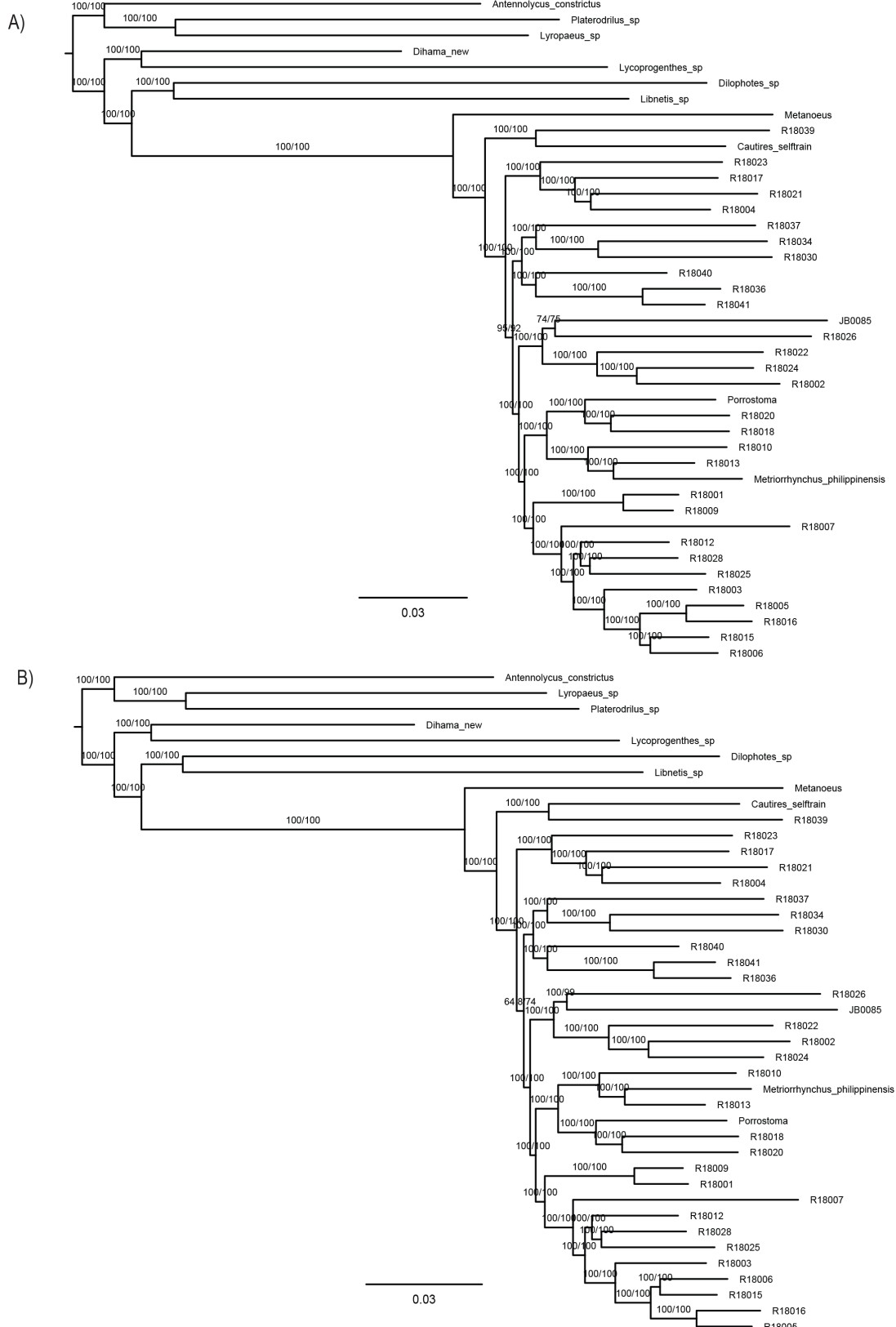

**Appendix 1—figure 2.** Maximum likelihood trees from IQ-TREE amino acid analysis. (**A**) Analysis of the dataset A-4109-AA with optimization of the partitioning scheme and (**B**) analysis of the dataset D-3370-AA_Bacoca with partitioning by gene. The depicted branch support values represent SH-aLRT and ultrafast bootstrap.

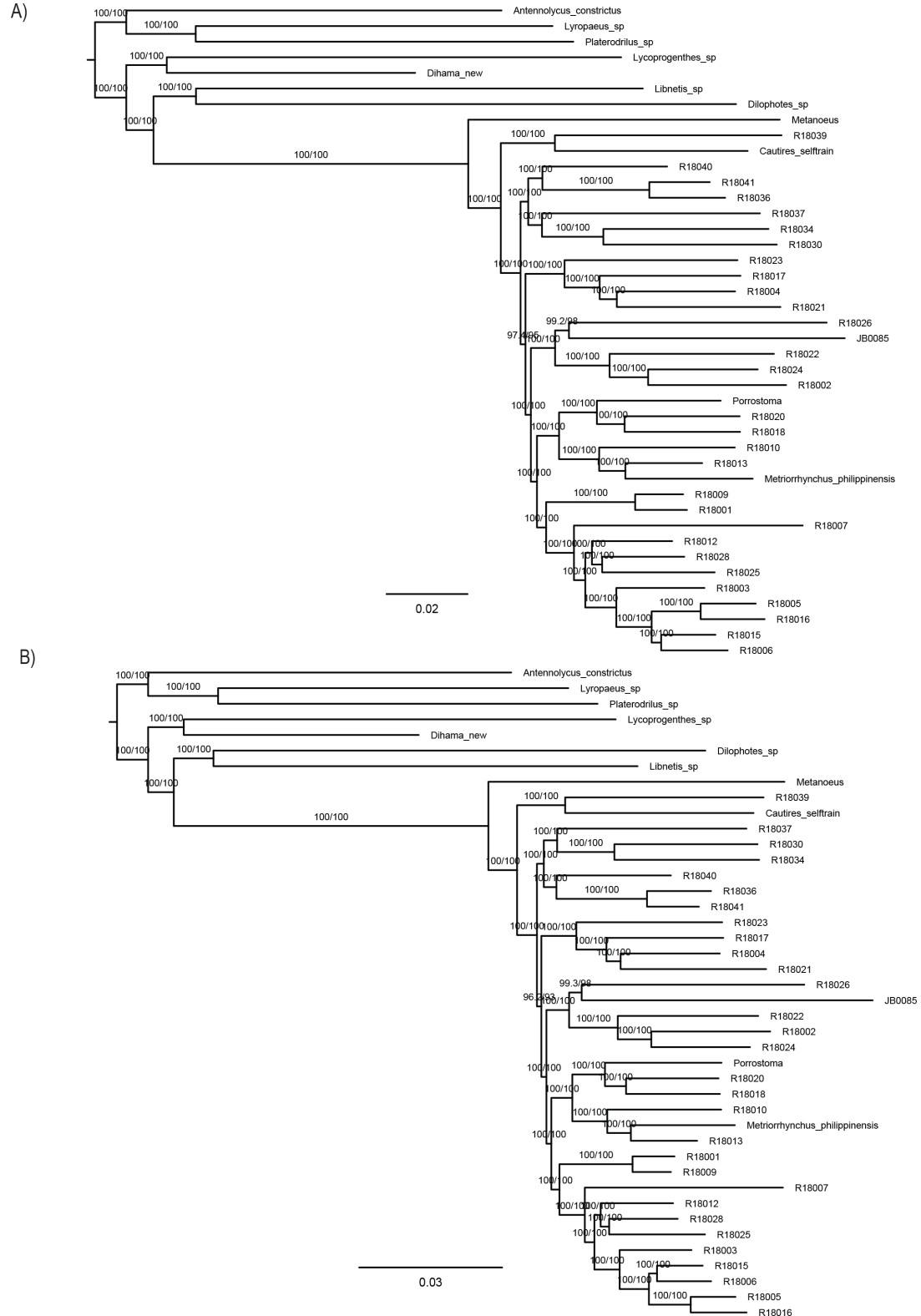

**Appendix 1—figure 3.** Maximum likelihood trees from IQ-TREE amino acid analysis. (**A**) analysis of the dataset A –F-1490-AA_Bacoca_decisive and (B)analysis of the dataset J-2129-AA_MARE. Both datasets were partitioned by gene. The depicted branch support values represent SH-aLRT and ultrafast bootstrap.

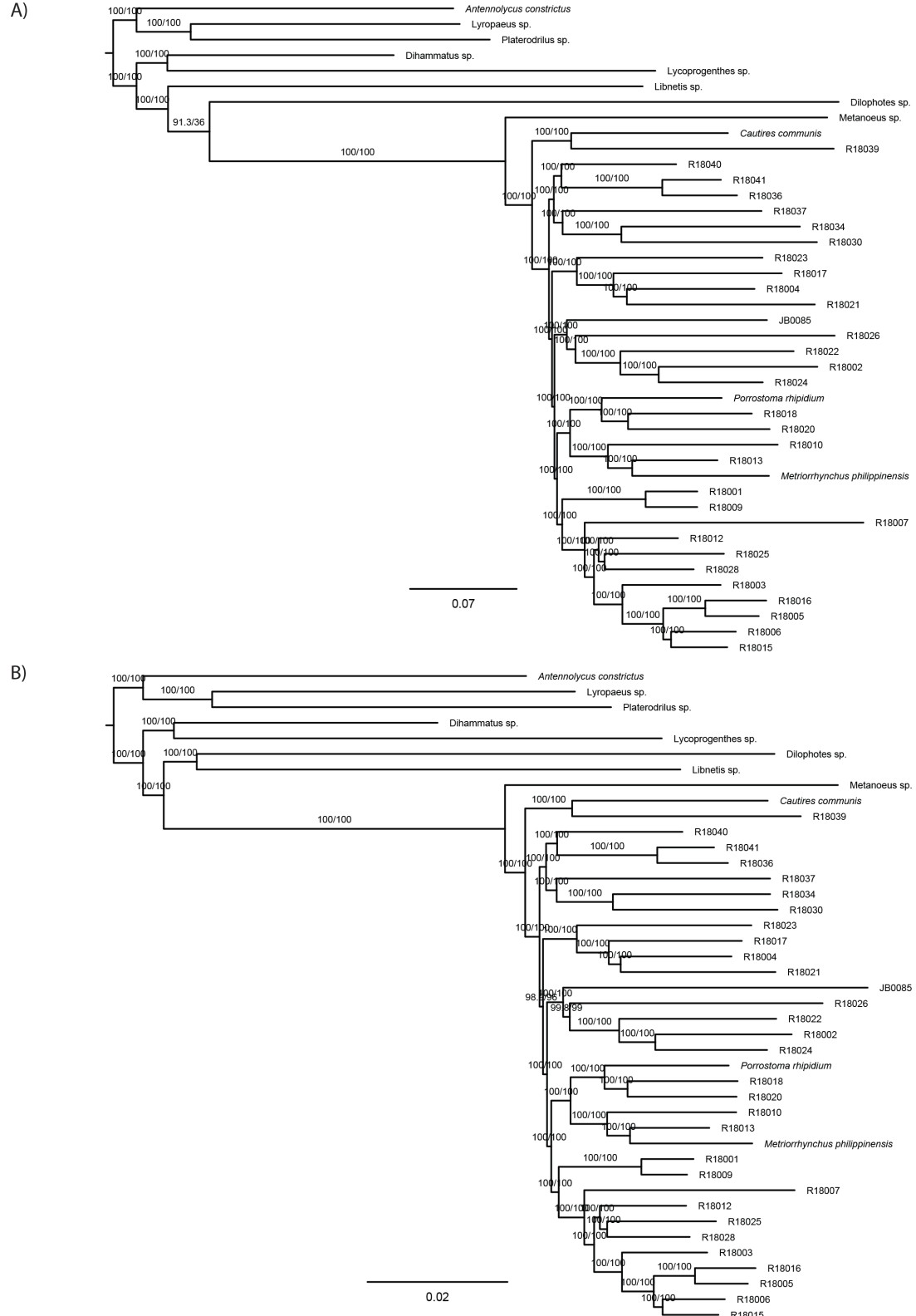

**Appendix 1—figure 4.** Maximum likelihood trees from IQ-TREE nucleotide analysis. (**A**) Analysis of the dataset B-4109-NT and (B)analysis of the dataset E-4109-NT2 using only second codon positions. Both datasets were partitioned by gene. The depicted branch support values represent SH-aLRT and ultrafast bootstrap.

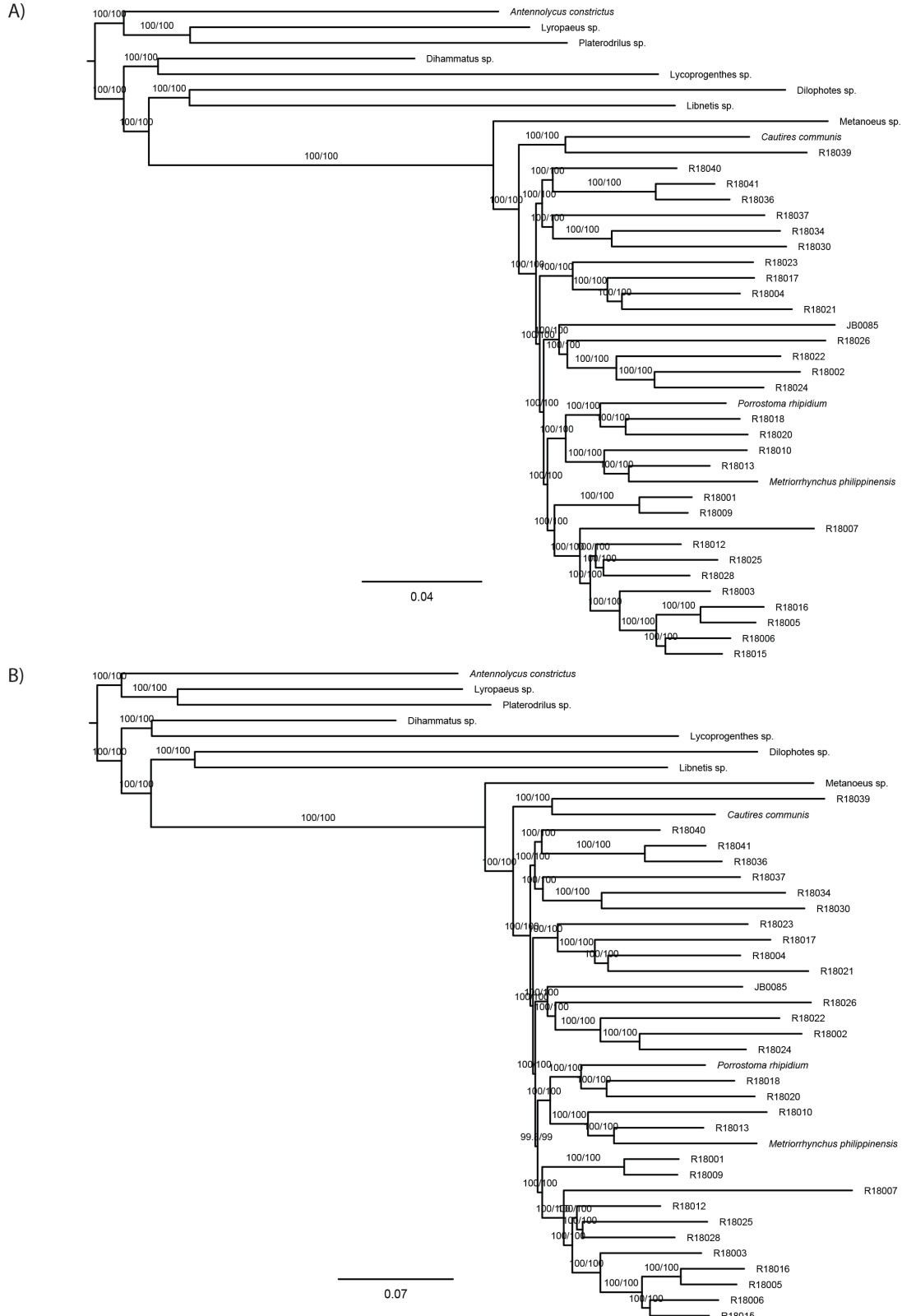

**Appendix 1—figure 5.** Maximum likelihood trees from IQ-TREE nucleotide analysis. (**A**) analysis of the datasetC-4109-NT12 using codon positions 1 + 2 and (**B**) analysis of the dataset G-NT-1767_MaxSymTest. Both datasets were partitioned by gene. The depicted branch support values represent SH-aLRT and ultrafast bootstrap.

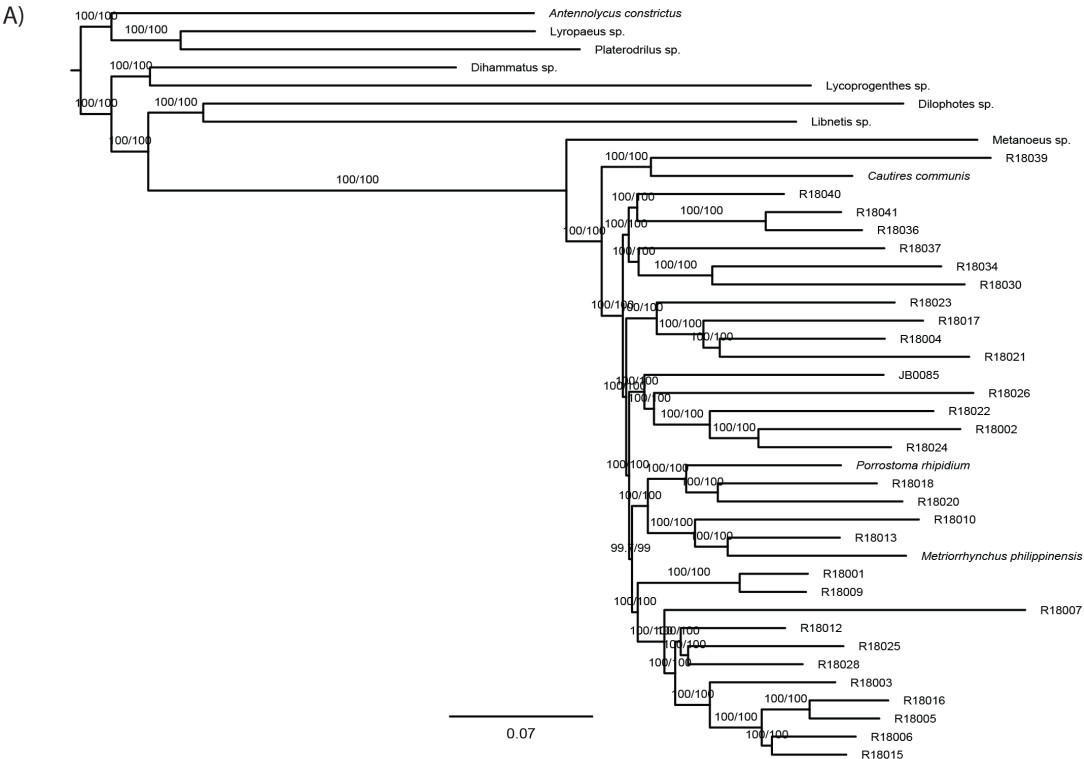

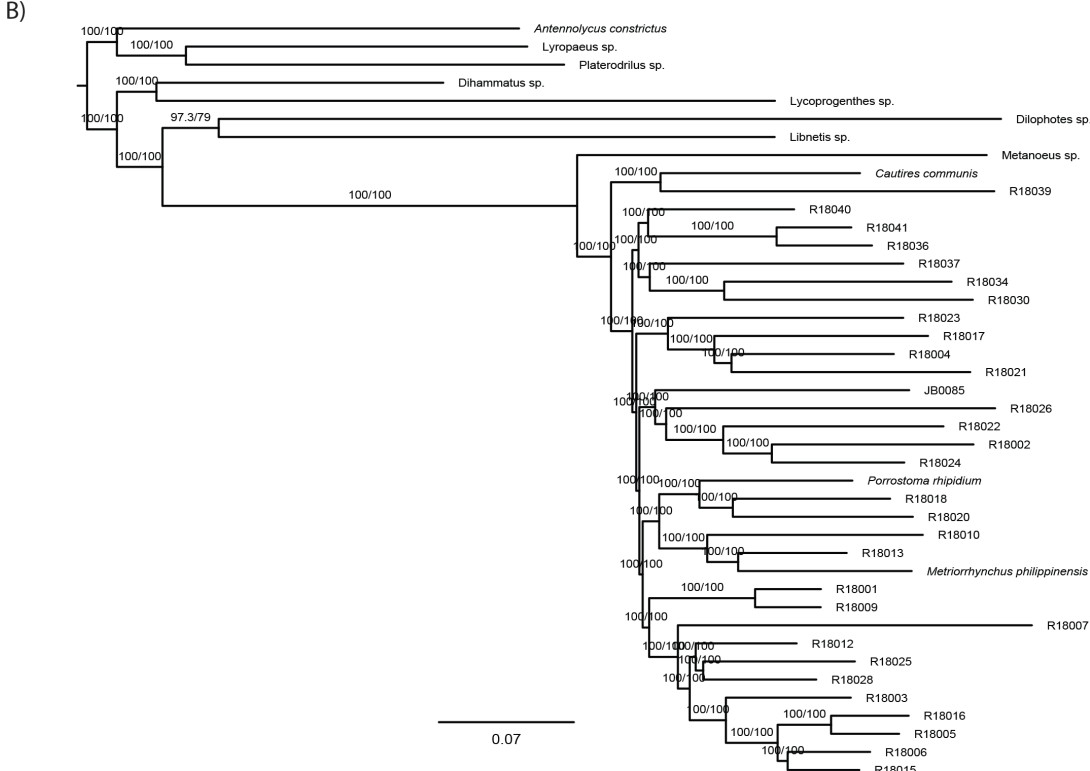

**Appendix 1—figure 6.** Maximum likelihood trees from IQ-TREE nucleotide analysis. (**A**) analysis of the dataset H-NT-1645_MaxSymTestmarginal and (**B**) analysis of the dataset I-NT-3905_MaxSymTestInternal. Both datasets were partitioned by gene. The depicted branch support values represent SH-aLRT and ultrafast bootstrap.

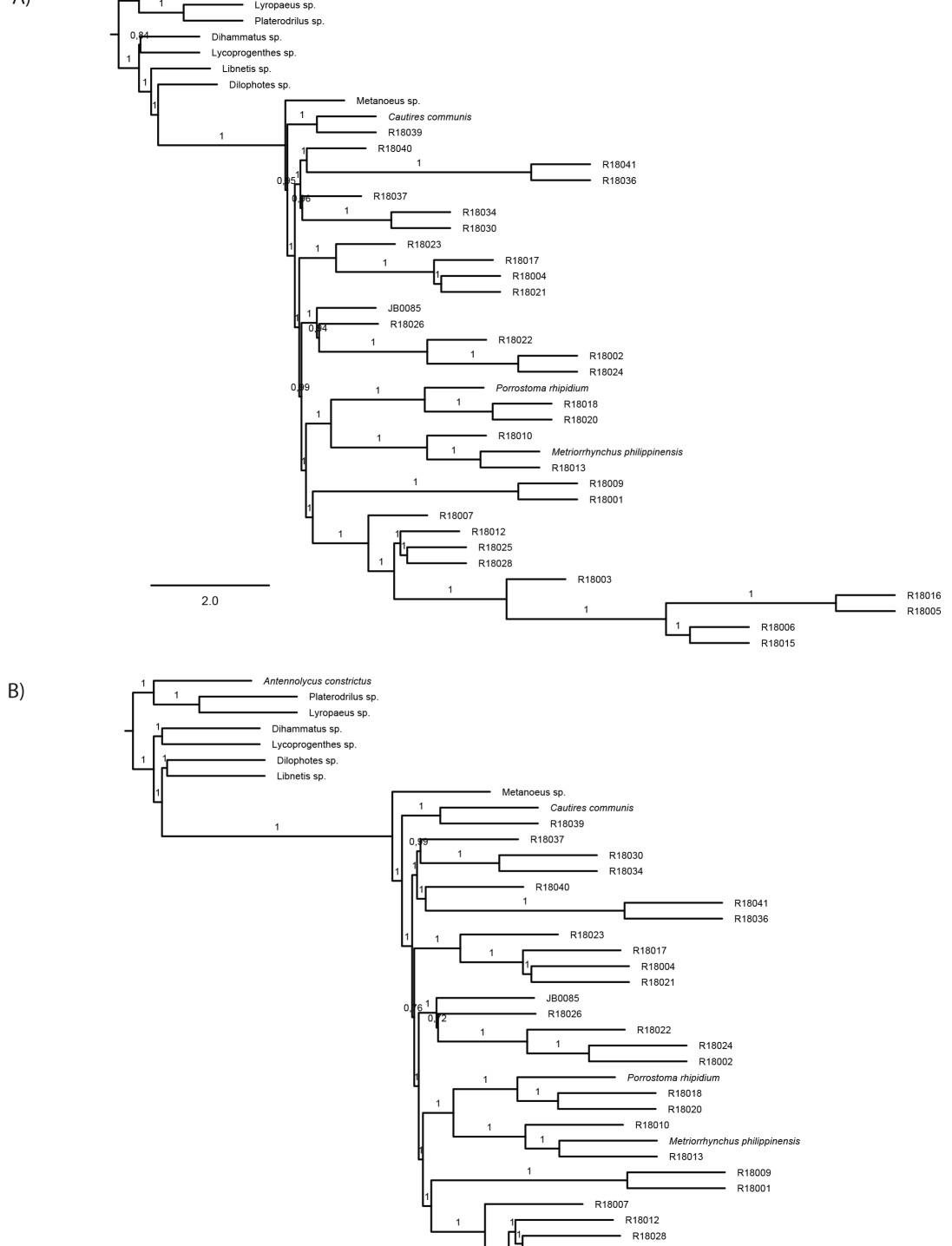

**Appendix 1—figure 7.** Topologies recovered by Astral analyses. (**A**) ASTRAL species trees with branch lengths in coalescent units as resulted from the analyses of individual IQ-TREE maximum likelihood gene trees of nucleotide dataset B-4109-NT and (B) ASTRAL species trees with branch lengths in coalescent units as resulted from the analyses of individual IQ-TREE maximum likelihood gene trees of amino acid dataset A-4199-AA. Numbers on nodes show local posterior probabilities (pp1). Quartet support for the alternative topologies (q1, q2, and q3), total number of induced quartet trees in the gene trees that support the alternative topologies (f1, f2, f3) and local posterior probabilities (pp1, pp2, pp3) are available at Dryad repository.

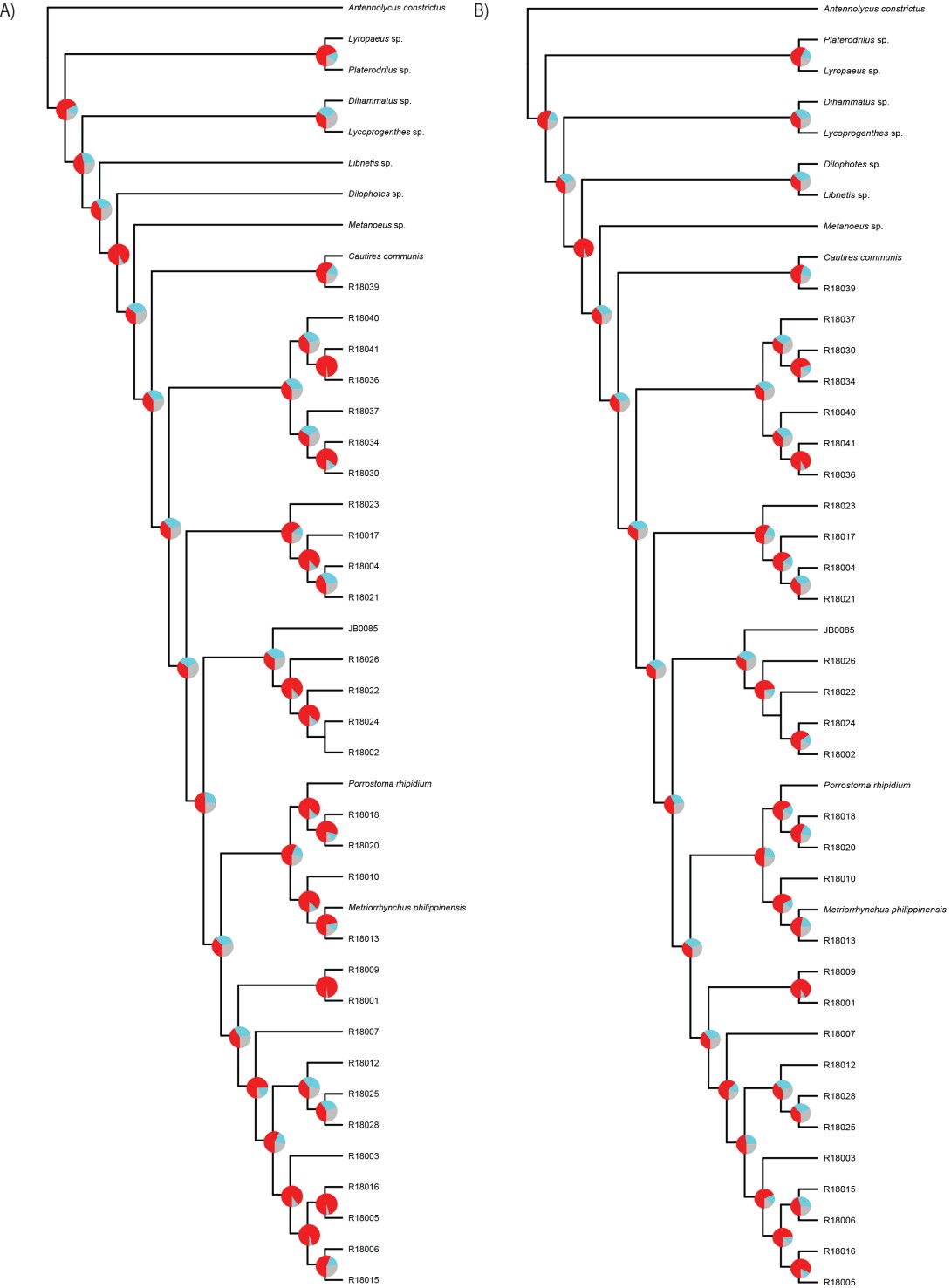

**Appendix 1—figure 8.** Phylogenomic topologies. (**A**) Phylogenetic relationships of Metriorrhynchini, resulted from the summary coalescent phylogenetic analysis with ASTRAL, when analyzing the full set of gene trees (4109 gene trees) inferred at the nucleotide level. (**B**) doitto at amino acids level. Pie charts on branches show ASTRAL quartet supports q1,q2,q3 (quartet-based frequencies of alternative quadripartition topologies around a given internode).Tree topologies correspond with *Appendix 1—figure 7*.

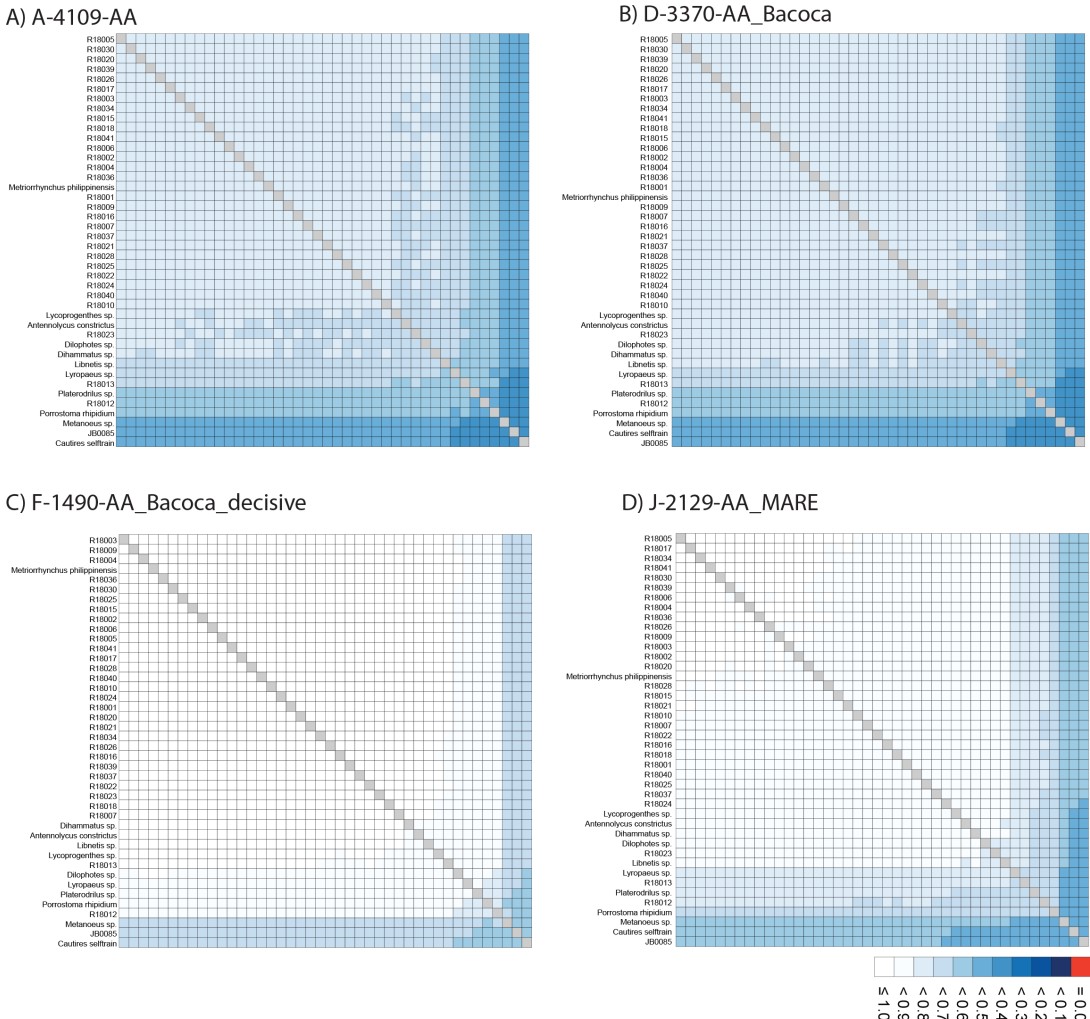

**Appendix 1—figure 9.** Alistat heatmaps. (**A–D**) AliStat rectangular heatmaps showing pairwise alignment completeness scores for all species included in the analyzed amino acid supermatrices. The abbreviations of datasets correspond to *Appendix 1—table 4*. Values closer to one indicate higher completeness scores for the pairwise sequence comparisons.

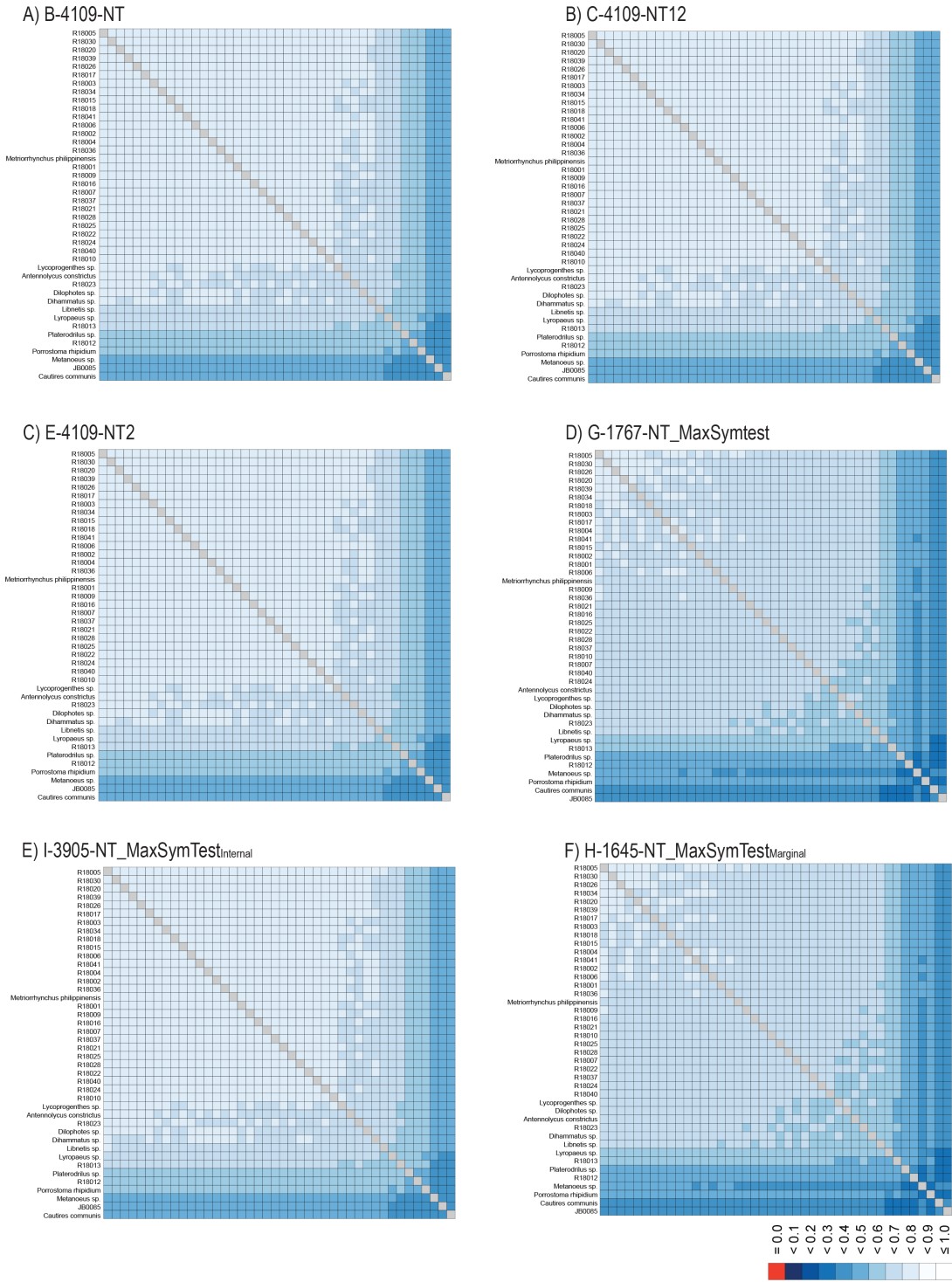

**Appendix 1—figure 10.** AliStat heatmaps. (**A–F**) AliStat rectangular heatmaps showing pairwise alignment completeness scores for all species included in the analyzed nucleotide supermatrices. The abbreviations of datasets correspond to *Appendix 1—table 4*. Values closer to one indicate higher completeness scores for the pairwise sequence comparisons.

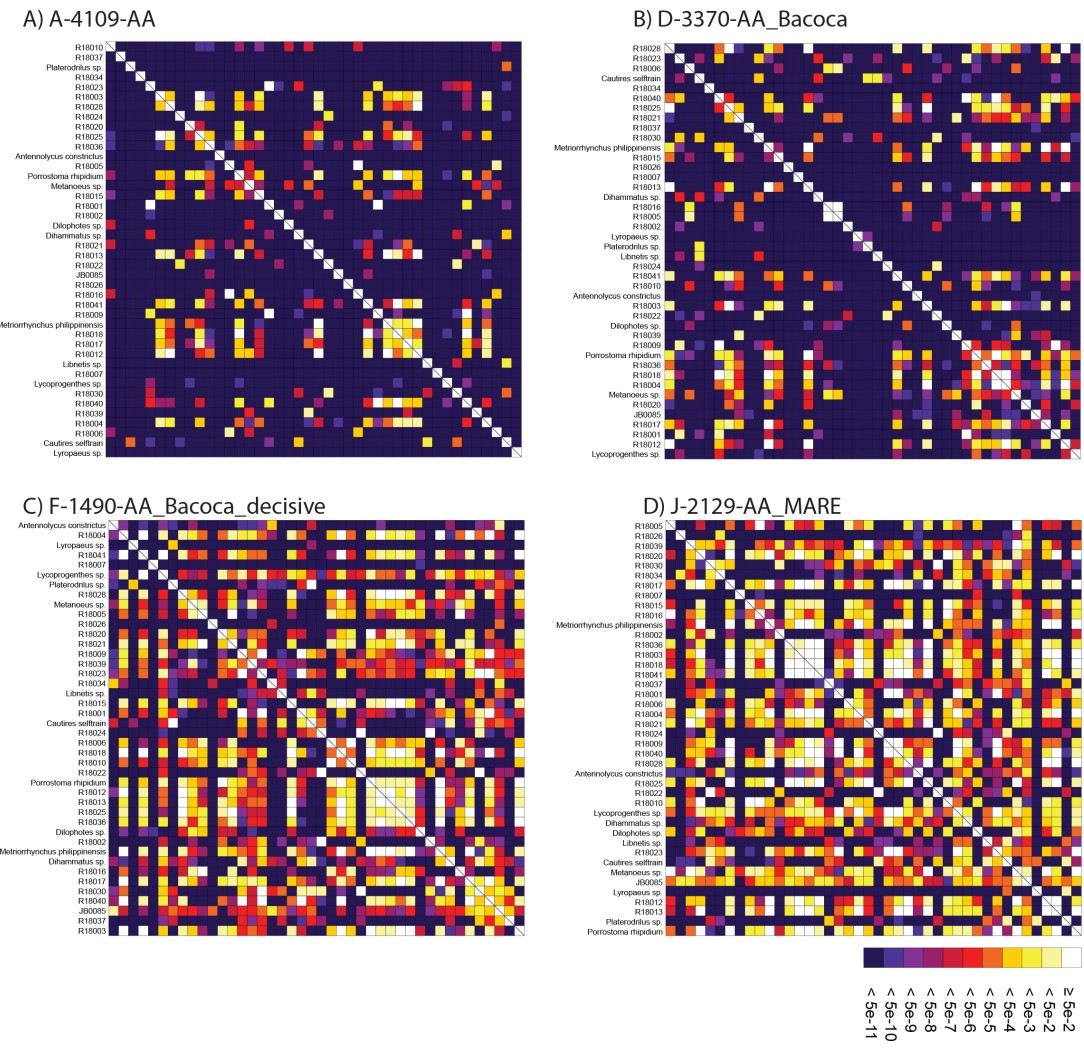

**Appendix 1—figure 11.** SymTest analyses. (**A–D**) Rectangular heatmap calculated with SymTest showing p-values for the pairwise Bowker's tests in the analyzed amino acid supermatrices. Darker boxes indicate lower p-values and thus larger deviation from evolution under SRH conditions.

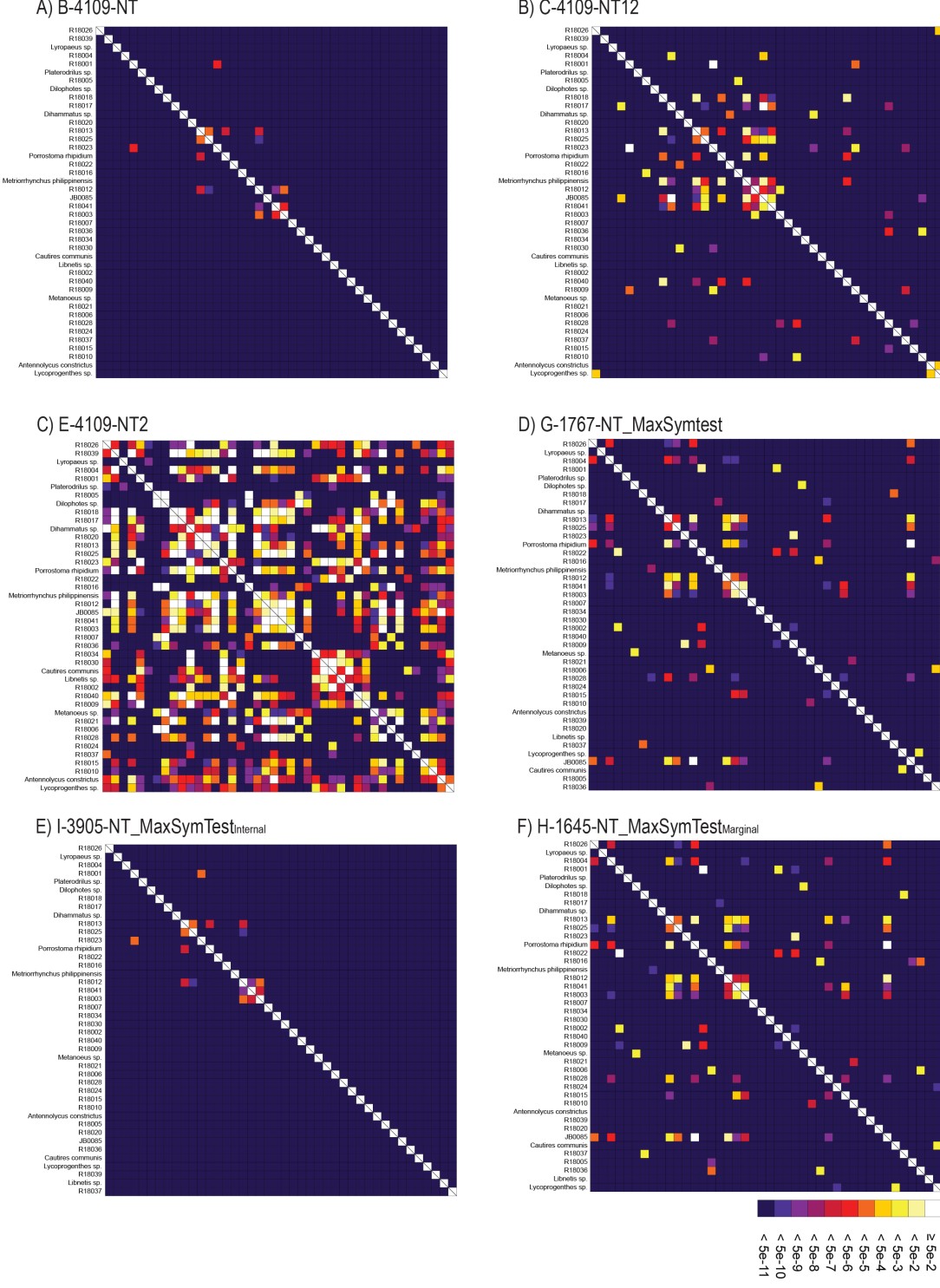

**Appendix 1—figure 12.** SymTest analyses. (**A–F**) Rectangular heat map calculated with SymTest showing p-values for the pairwise Bowker's tests in the analyzed nucleotide supermatrices. Darker boxes indicate lower p-values and thus larger deviation from the evolution under SRH conditions.

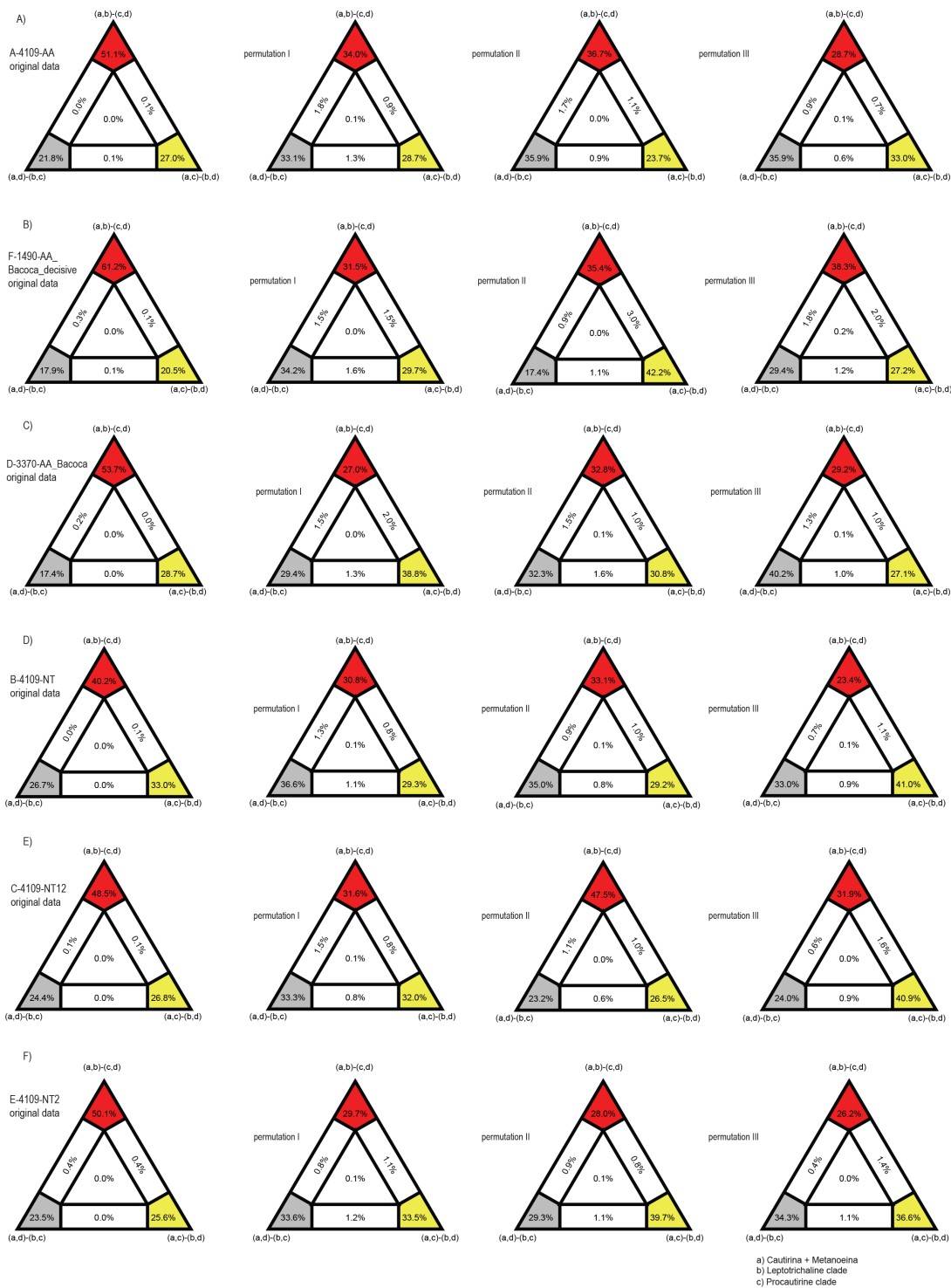

**Appendix 1—figure 13.** Results of FcLM analyses testing alternative phylogenetic hypotheses about placement of procautirine and leptotrichaline clades applied for various supermatrices. The first column shows the results of FcLM when the original data were analyzed. The second column shows the results of FcLM after the phylogenetic signal had been eliminated from data. The third column show the results of FcLM after elimination of the phylogenetic signal and inhomogeneous amino-acid or nucleotide composition. The fourth column show the results of FcLM after the elimination of phylogenetic signal, inhomogeneous amino-acid or nucleotide composition and with randomized data coverage within all meta-partitions (see Supplementary methods for details).

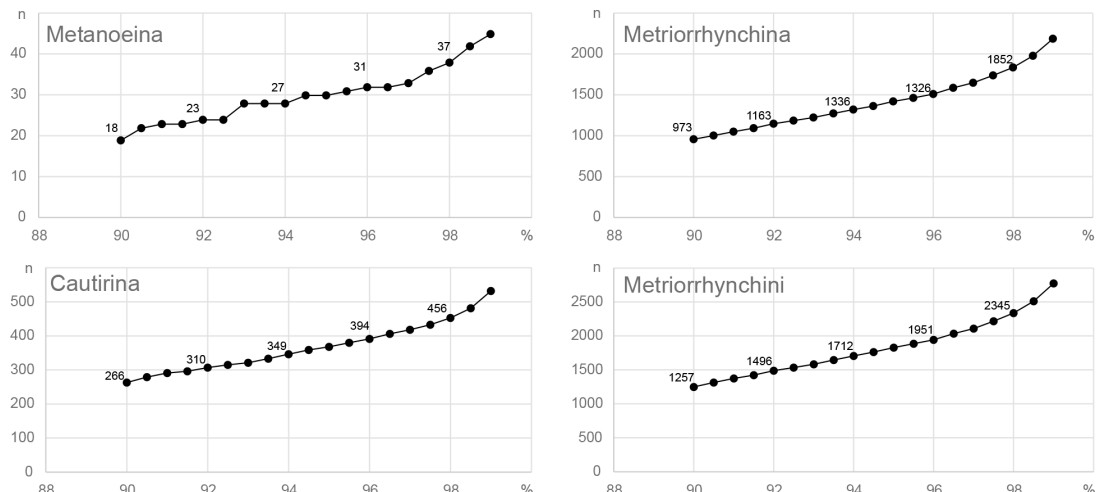

**Appendix 1—figure 14.** Tribal and subtribal mOTUs delimitation using CD-hit-est.
Axis X represent the number of operational taxonomic units, whereas axis Y shows the delimitation threshold.

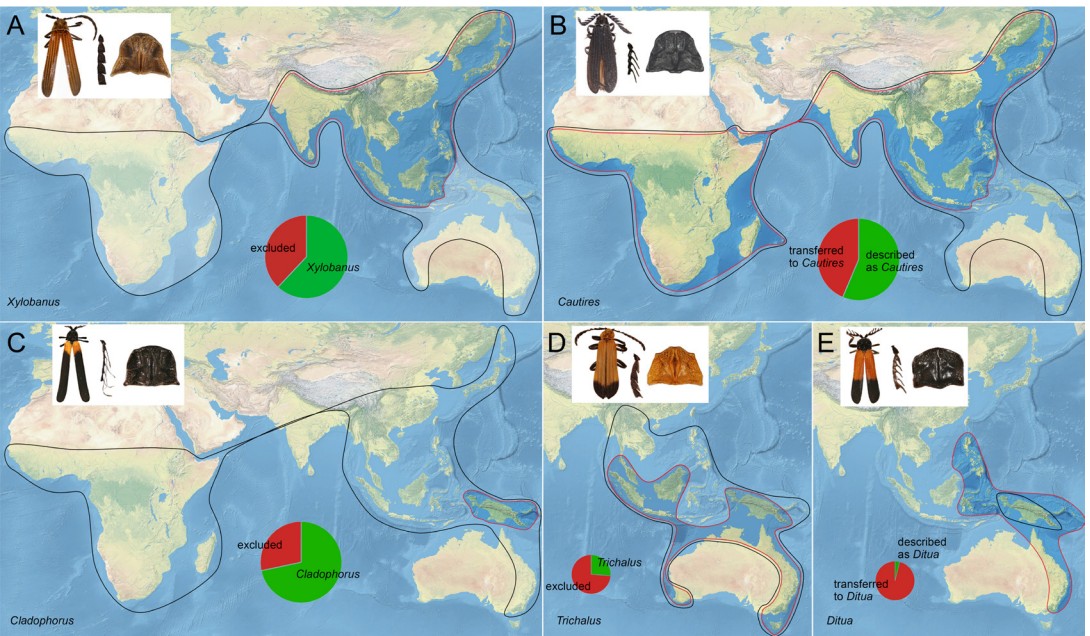

**Appendix 1—figure 15.** Distribution of selected Metriorrhynchini genera (originally assumed and revised).

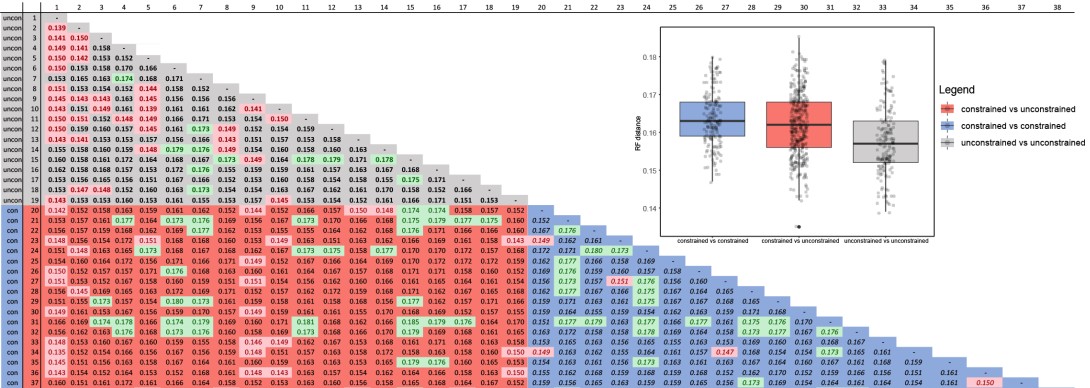

**Appendix 1—figure 16.** Comparison of the Robinson-Fould (RF) distances among tree searches with constrained topology and tree searches with unconstrained topology. Green values represent top 10% RF- distances, and red values shows bottom 10% RF- distances. The blue part of the graph represents constrained trees values, the grey part of the graph represents unconstrained trees values, the red part of the graph represents constrained/ unconstrained trees values.

**Appendix 1—table 1.** Detailed information on regional sampling.

| | Metriorrhynchina | Cautirina | Metanoeina | Metriorrhynchini |
|---|---|---|---|---|
| Region | # of specimens | # of specimens | # of specimens | # of specimens |
| **Australian region** | 4489 | | | 4489 |
| Australian continent | 3964 | | | 3964 |
| Australia | 475 | | | 475 |
| *New South Wales* | 64 | | | 64 |
| *Northern Territory* | 18 | | | 18 |
| *Queensland* | 382 | | | 382 |
| *South Australia* | 1 | | | 1 |
| *Western Australia* | 10 | | | 10 |
| New Zealand | 1 | | | 1 |
| New Guinea | 3461 | | | 3461 |
| Solomon Islands | 19 | | | 19 |
| Aru Islands | 8 | | | 8 |
| | | | | |
| **Wallacea** | 525 | 15 | | 540 |
| Maluku-Buru | 15 | | | 15 |
| Halmahera | 169 | | | 169 |
| Sulawesi | 341 | 15 | | |
| | | | | |
| Oriental region | 278 | 1,311 | 113 | 1,692 |
| **Philippines** | 48 | 37 | 14 | 99 |
| *Mindanao* | 11 | 3 | 9 | 23 |
| *Negros* | 2 | | | |
| *Palawan* | 29 | 34 | 5 | 68 |
| *Sibuyan* | 4 | | | 4 |
| *Luzon* | 2 | | | 2 |

*Appendix 1—table 1 Continued on next page*

*Appendix 1—table 1 Continued*

| | Metriorrhynchina | Cautirina | Metanoeina | Metriorrhynchini |
|---|---|---|---|---|
| Continental Asia | **230** | 1264 | **99** | 1593 |
| Malaya | 34 | 270 | 8 | 348 |
| Java | 21 | 45 | 6 | 72 |
| Bali | 2 | 1 | | 3 |
| Sumatra | 56 | 201 | 26 | 283 |
| Borneo | 82 | 253 | 22 | 357 |
| Cambodia | 7 | 13 | | 20 |
| Indo-Burma | 23 | 145 | 10 | 178 |
| China incl. Taiwan | 5 | 100 | 23 | 128 |
| India | | 59 | | 59 |
| Japan | | 177 | 4 | 181 |
| | | | | |
| Afrotropical region | | 233 | | 233 |
| Africa | | 175 | | 175 |
| Madagascar | | 58 | | 58 |
| | | | | |
| Total | 4767 | 1549 | **113** | 6429 |

**Appendix 1—table 2.** List of material for phylogenomic analyses.

**Ingroup**

| Voucher | Species | | Clade | Geographic origin | Data source | Data type | Tissue type |
|---|---|---|---|---|---|---|---|
| R18010 | Metriorrhynchus s. l. | Metriorrhynchina | Porrostomine | New Guinea | This study | ILLUMINA RNA-seq PE-reads | Whole animal |
| R18013 | Metriorrhynchus sp. | Metriorrhynchina | Porrostomine | New Guinea | This study | ILLUMINA RNA-seq PE-reads | Whole animal |
| R20001 | *Metriorrhynchus philippinensis* | Metriorrhynchina | Porrostomine | Philippines | This study | ILLUMINA RNA-seq PE-reads | Whole animal |
| R18018 | Metriorrhynchus s. l. | Metriorrhynchina | Porrostomine | New Guinea | This study | ILLUMINA RNA-seq PE-reads | Whole animal |
| R18020 | Metriorrhynchus s. l. | Metriorrhynchina | Porrostomine | New Guinea | This study | ILLUMINA RNA-seq PE-reads | Whole animal |
| 1_kite | *Porrostoma rhipidium* | Metriorrhynchina | Porrostomine | Australia | *McKenna et al., 2019* | ILLUMINA RNA-seq PE-reads | Whole animal |
| R18001 | Cladophorus sp. | Metriorrhynchina | Cladophorine | New Guinea | This study | ILLUMINA RNA-seq PE-reads | Whole animal |
| R18009 | Cladophorus sp. | Metriorrhynchina | Cladophorine | New Guinea | This study | ILLUMINA RNA-seq PE-reads | Whole animal |
| R18007 | Pseudodontocerus sp. | Metriorrhynchina | Cladophorine | New Guinea | This study | ILLUMINA RNA-seq PE-reads | Whole animal |
| R18012 | Ditua s. l. | Metriorrhynchina | Cladophorine | New Guinea | This study | ILLUMINA RNA-seq PE-reads | Whole animal |
| R18025 | Ditua s. l. | Metriorrhynchina | Cladophorine | New Guinea | This study | ILLUMINA RNA-seq PE-reads | Whole animal |
| R18028 | Ditua s. l. | Metriorrhynchina | Cladophorine | New Guinea | This study | ILLUMINA RNA-seq PE-reads | Whole animal |
| R18003 | Ditua s. l. | Metriorrhynchina | Cladophorine | New Guinea | This study | ILLUMINA RNA-seq PE-reads | Whole animal |
| R18016 | Ditua s. l. | Metriorrhynchina | Cladophorine | New Guinea | This study | ILLUMINA RNA-seq PE-reads | Whole animal |
| R18005 | Ditua s. l. | Metriorrhynchina | Cladophorine | New Guinea | This study | ILLUMINA RNA-seq PE-reads | Whole animal |
| R18015 | Ditua s. l. | Metriorrhynchina | Cladophorine | New Guinea | This study | ILLUMINA RNA-seq PE-reads | Whole animal |
| R18006 | Ditua s. l. | Metriorrhynchina | Cladophorine | New Guinea | This study | ILLUMINA RNA-seq PE-reads | Whole animal |
| R18002 | Microtrichalus sp. | Metriorrhynchina | Trichaline | New Guinea | This study | ILLUMINA RNA-seq PE-reads | Whole animal |
| R18024 | Eniclases sp. | Metriorrhynchina | Trichaline | New Guinea | This study | ILLUMINA RNA-seq PE-reads | Whole animal |

*Appendix 1—table 2 Continued on next page*

*Appendix 1—table 2 Continued*

**Ingroup**

| Voucher | Species | | Clade | Geographic origin | Data source | Data type | Tissue type |
|---|---|---|---|---|---|---|---|
| R18022 | Diatrichalus sp. | Metriorrhynchina | Trichaline | New Guinea | This study | ILLUMINA RNA-seq PE-reads | Whole animal |
| R18026 | Trichaline sp. | Metriorrhynchina | Trichaline | New Guinea | This study | ILLUMINA RNA-seq PE-reads | Whole animal |
| JB0085 | Trichaline sp. | Metriorrhynchina | Trichaline | New Guinea | This study | ILLUMINA WGS-seq PE-reads | Thoracic muscles |
| R18004 | Procautires sp. | Metriorrhynchina | Procautirine | New Guinea | This study | ILLUMINA RNA-seq PE-reads | Whole animal |
| R18017 | Procautires sp. | Metriorrhynchina | Procautirine | New Guinea | This study | ILLUMINA RNA-seq PE-reads | Whole animal |
| R18021 | Procautires sp. | Metriorrhynchina | Procautirine | New Guinea | This study | ILLUMINA RNA-seq PE-reads | Whole animal |
| R18023 | Procautires sp. | Metriorrhynchina | Procautirine | New Guinea | This study | ILLUMINA RNA-seq PE-reads | Whole animal |
| R18037 | *Broxylus sp.* | Metriorrhynchina | Leptotrichaline | Sulawesi | This study | ILLUMINA RNA-seq PE-reads | Whole animal |
| R18030 | Leptotrichalus sp. | Metriorrhynchina | Leptotrichaline | Sulawesi | This study | ILLUMINA RNA-seq PE-reads | Whole animal |
| R18034 | Wakarumbia sp. | Metriorrhynchina | Leptotrichaline | Sulawesi | This study | ILLUMINA RNA-seq PE-reads | Whole animal |
| R18036 | *Sulabanus sp.* | Metriorrhynchina | Leptotrichaline | Sulawesi | This study | ILLUMINA RNA-seq PE-reads | Whole animal |
| R18041 | *Sulabanus sp.* | Metriorrhynchina | Leptotrichaline | Sulawesi | This study | ILLUMINA RNA-seq PE-reads | Whole animal |
| R18040 | *Mangkutanus sp.* | Metriorrhynchina | Leptotrichaline | Sulawesi | *Kusy et al., 2019* | ILLUMINA RNA-seq PE-reads | Whole animal |
| R18039 | Xylobanus sp. | Cautirina | | Sulawesi | This study | ILLUMINA RNA-seq PE-reads | Whole animal |
| AJ0013 | *Cautires communis* | Cautirina | | Malaysia | *Kusy et al., 2019* | ILLUMINA WGS-seq PE-reads | Thoracic muscles |
| G19002 | Metanoeus sp. | Metanoeina | | Malaysia | This study | ILLUMINA WGS-seq PE-reads | Thoracic muscles |

**Outgroup**

| | Species | | | Geographic origin | Data source | Data type | Tissue type |
|---|---|---|---|---|---|---|---|
| | Dilophotes sp. | | | Malaysia | *Kusy et al., 2019* | ILLUMINA RNA-seq PE-reads | Whole animal |
| | Dihammatus sp. | | | Malaysia | *Kusy et al., 2019* | ILLUMINA RNA-seq PE-reads | Whole animal |
| | *Lycoprogentes sp.* | | | Malaysia | *Kusy et al., 2019* | ILLUMINA RNA-seq PE-reads | Whole animal |
| | *Libnetis sp.* | | | Malaysia | *Kusy et al., 2019* | ILLUMINA RNA-seq PE-reads | Whole animal |
| | Platerodrilus sp. | | | Malaysia | *Kusy et al., 2019* | ILLUMINA RNA-seq PE-reads | Whole animal |
| | *Lyropaeus optabilis* | | | Malaysia | *Kusy et al., 2019* | ILLUMINA RNA-seq PE-reads | Whole animal |
| | *Antennolycus constrictus* | | | Malaysia | *Kusy et al., 2019* | ILLUMINA RNA-seq PE-reads | Whole animal |

**Appendix 1—table 3.** Overview of official gene sets of six reference species used for ortholog assessment.
Number of genes correspond with OrthoDB 9.1.

| Species | Accession | # of contigs | Source | Download date | Reference |
|---|---|---|---|---|---|
| Onthophagus taurus | PRJNA167478 | 17,483 | i5K | 05.03.2017 | 1 |
| Tribolium castaneum | PRJNA12540 | 16,631 | iBeetle | 05.03.2017 | 2, 3 |
| Dendroctonus ponderosae | PRJNA360270 | 13,088 | ENS Metazoa | 05.03.2017 | 4 |
| Anoplophora glabripennis | PRJNA167479 | 22,035 | i5K | 05.03.2017 | 5 |
| Leptinotarsa decemlineata | PRJNA171749 | 24,671 | i5K | 05.03.2017 | 1 |
| Agrilus planipennis | PRJNA230921 | 15,497 | i5K | 05.03.2017 | 1 |

**Appendix 1—table 4.** Detailed information and statistics of each generated dataset.

| Dataset name | Number of taxa | Number of partitions | Number of alignment sites | Completeness score (Ca) AliStat | Percentage of pairwise P-values < 0.05 for the Bowker's test | (SV) MARE Matrix saturation | (IC) MARE Information content | Partition scheme optimalization |
|---|---|---|---|---|---|---|---|---|
| **Amino acids** | | | | | | | | |
| A-4109-AA | 42 | 4,109 | 1892691 | 0.772127 | 98.61% | 0.888 | 0.506 | Yes |
| D-3370-AA_Bacoca | 42 | 3,370 | 1672362 | 0.780363 | 97.91% | 0.880 | 0.511 | |
| F-1490-AA_Bacoca_decisive | 42 | 1,490 | 673,102 | 0.925082 | 94.08% | 1 | 0.594 | |
| J-2129-AA_MARE | 42 | 2,129 | 959,741 | 0.876331 | 90.82% | 0.964 | 0.648 | |
| **Nucleoides** | | | | | | | | |
| B-4109-NT | 42 | 4,109 | 5678073 | 0.772133 | 100% | NA | NA | |
| C-4109-NT12 | 42 | 4,109 | 3785382 | 0.772133 | 99.77% | NA | NA | |
| E-4109-NT2 | 42 | 4,109 | 1892691 | '0.772134 | 92.45% | NA | NA | |
| G-NT-1767_MaxSymTest | 42 | 1,767 | 2413164 | '0.713912 | 99.88% | NA | NA | |
| H-NT-1645_MaxSymTestmarginal | 42 | 1,645 | 2233485 | '0.712208 | 99.77% | NA | NA | |
| I-NT-3905_MaxSymTestInternal | 42 | 3,905 | 5377449 | '0.771645 | 100% | NA | NA | |

**Appendix 1—table 5.** Characteristics of concatenated super-matrices (mitochondrial fragments) and used models of the DNA evolution (descriptions of columns abbreviation listed bellow tables). Abbreviations. Seq – Number of sequences, Site – Number of bases, Infor – Number of parsimony-informative sites, Invar – Number of invariant sites, Model – Best-fit model according to BIC using ModelFinder.

| Subset | Seqs | Sites | Infor | Invar | Model |
|---|---|---|---|---|---|
| **Complete dataset** | | | | | |
| nad5 | 3,205 | 987 | 809 | 88 | GTR + F + I + G4 |
| cox | 5,935 | 1,068 | 812 | 134 | GTR + F + I + G4 |
| 16 s | 2,381 | 875 | 577 | 201 | GTR + F + I + G4 |
| **98% OTUs delimitation dataset** | | | | | |
| nad5 | 1,252 | 969 | 776 | 115 | GTR + F + I + G4 |
| tRNA Leu | 2,245 | 62 | 34 | 17 | GTR + F + G4 |
| cox1 | 2,395 | 783 | 536 | 141 | GTR + F + I + G4 |
| cox2 | 2,192 | 216 | 194 | 11 | GTR + F + I + G4 |
| 16 s | 1,115 | 834 | 512 | 237 | GTR + F + I + G4 |

