## [Editor Report]

This manuscript provides clear ideas regarding the usage of next-generation sequencing data, and of more traditional mtDNA markers, to rapidly increase biodiversity inventories. You demonstrate how biodiversity information analyses done in the Metriorrhynchini, a hyperdiverse tropical insect group, can be rapidly expanded via targeted field research and large-scale sequencing. The study sets a benchmark for the spatiotemporal evaluation of tropical biodiversity, supports evidence-based conservation planning, and provides a robust framework for systematic, biogeographic, and evolutionary studies.

---

## [Decision Letter]

**Decision letter after peer review:**

Thank you for submitting your article "Phylogenomic and mitogenomic data can accelerate inventorying of tropical beetles during the current biodiversity crisis" for consideration by *eLife*. Your article has been reviewed by 2 peer reviewers, and the evaluation has been overseen by a Reviewing Editor and Detlef Weigel as the Senior Editor. The following individual involved in review of your submission has agreed to reveal their identity: Ming Bai (Reviewer #3).

Essential revisions:

The reviewers agree that the authors did a nice sampling and produce an interesting inventory, but that the authors need to be cautious in their claims (as noted in their results or this "new" methodology). One major concern is with the validity of the premise that this paper helps to accelerate, or provides a framework that accelerates tropical insect inventories. For example, it will be very important if the authors develop the 4-step procedure (Lines 90-97) they use to justify the importance of the work. What is unique (or new) in their methodology, that can accelerate inventory assembly? Despite Figure 5, which establish differences between their methods and basic Sanger sequencing, the reviewers are not yet convinced that this paper really moves the field forward. Alternatively, the authors can temper the *acceleration biodiscovery* framework, and provide a more robust background on how Illumina data may be essential for obtaining a backbone where to put the Sanger data, and how this is not frequently used yet.

*Reviewer #2 (Recommendations for the authors):*

Overall, I like the manuscript and I think it can be published after some modifications. The most important point is the claims about mOTUs/putative species. As a scientist, we should be cautious when interpreting our data and especially when selecting different parameters (like the threshold). Authors should not provide just their most optimistic (or most striking) estimates and should include different estimates and consider all of them over the manuscript (from abstract to conclusions).

Also, It is surprising the amount of text used for the phylogenomic analyses, it is a very appropriate methodological part but barely mentioned in the results. Authors should improve the result/discussion of this part, describing and comparing what they produce with other studies (if any).

Finally, the use of "person-months of focused field research" should be considered to be removed, as it did not provide any results or comparison with other study groups (which won't be possible).

*Reviewer #3 (Recommendations for the authors):*

Conservation efforts must be evidence-based, so rapid and economically feasible methods should be used to quantify diversity and distribution patterns. The principal objective of this study is to demonstrate how biodiversity information for a hyperdiverse tropical group can be rapidly expanded via targeted field research and large-scale sequencing. The authors have attempted to overcome current impediments to the gathering of biodiversity data by using integrative phylogenomic and three mtDNA fragment analyses. As a model, they sequenced the Metriorrhynchini beetle fauna, sampled from ~700 localities in three continents. The species-rich dataset included ~6,500 terminals, >2,300 putative species, more than a half of them unknown to science. It is an amazing finding. Their information and phylogenetic hypotheses can be a resource for higher-level phylogenetics, population genetics, phylogeographic studies, and biodiversity estimation. At the same time, they want to show how limited the taxonomical knowledge is and how this lack is hindering biodiversity research and management. It is a nice and well-designed study with a very important meaning. I would be happy to recommend it to be accepted after the revision. However before it is formally accepted, there are several questions followed:

1. The definition of species is very complex. The molecular species, e.g. OTU, is still different from morphological species using traditional taxonomic methods. Of course, it is very common to use molecular species to define and infer biodiversity in many studies. For this study, ideally, the authors can express their comments on the relationships between molecular species and morphological species. For example, Sharkey et al. (2021 Zookeys) published 403 new species only with molecular information.

2. P8L199: The constituent clades should be point out based on which tree. Pruned mitogenomic tree with and without constraints or Constrained mitogenomics?

3. P8L205-208: How many mOTUs if using thresholds of 2%? 2345 or 2356? It is inconsistent in this paragraph.

4. P10L244: This table is not too clearly to explain shared regions. If illustrate the meanings of number before and after "/", it will be better.

5. P14L369: What's the references?

6. P16L448: This discovery should be shown in Results part.

7. P18L528: What is the voucher number of Metanoeus sp.?

8. P96 S18: The figure S18 is unclear. It should be replaced by clear one.

---

## [Author Response]

Essential revisions:The reviewers agree that the authors did a nice sampling and produce an interesting inventory, but that the authors need to be cautious in their claims (as noted in their results or this "new" methodology). One major concern is with the validity of the premise that this paper helps to accelerate, or provides a framework that accelerates tropical insect inventories. For example, it will be very important if the authors develop the 4-step procedure (Lines 90-97) they use to justify the importance of the work. What is unique (or new) in their methodology, that can accelerate inventory assembly? Despite Figure 5, which establish differences between their methods and basic Sanger sequencing, the reviewers are not yet convinced that this paper really moves the field forward. Alternatively, the authors can temper the acceleration biodiscovery framework, and provide a more robust background on how Illumina data may be essential for obtaining a backbone where to put the Sanger data, and how this is not frequently used yet.

Many thanks for your comments. We agree with your suggestions and tried to address them.

We have added further information on our approach. Principally, we combine phylogenomics (both transcriptomes and WGS data) that has been used for deep phylogenies like insect order relationships or the beetle classification with short DNA fragments that resolve only shallow relationships. We tried to limit the number of transcriptomes to avoid high costs. Further, we elaborated each dataset contributes to results. Separately, neither phylogenomics nor short fragment analyses provide necessary information. We are aware that even our approach is labour intensive, but we argue that reporting the presence of ~1,000 undescribed species by a single group for a tribe of beetles is in the clear contrast with the slow pace of traditional morphology based taxonomy (the journal Zootaxa published 30,000 articles with 60,000 new species in 20 years; Zhang, 2021). Additionally, as we work with preserved vouchers, our research can be used for further studies integrating morphology and molecular phylogenetics.

Below, we answer how specific comments by reviewers were addressed. We tried to follow the suggestions as closely as possible.

Reviewer #2 (Recommendations for the authors):Overall, I like the manuscript and I think it can be published after some modifications. The most important point is the claims about mOTUs/putative species. As a scientist, we should be cautious when interpreting our data and especially when selecting different parameters (like the threshold). Authors should not provide just their most optimistic (or most striking) estimates and should include different estimates and consider all of them over the manuscript (from abstract to conclusions).

Thanks for the comment. We modified the text accordingly, but we also discussed the lower threshold. We pointed more clearly to the Figure 17 where are shown numbers of putative species for 20 thresholds (1-10%, increased by 0.5%). Our preference for lower threshold has been based on the earlier published studies on Metriorrhynchini that have shown conspicuous morphological differentiation between New Guinean putative species which shared similar mtDNA (Kalousova and Bocak, 2017; Bocek et al., 2019). We studied the congruence of mtDNA/morphology species delimitation with nextRAD data in the later study. We proposed that morphological differentiation was driven by selection in different mimetic communities and by ecological shift (exemplified by possible shift to nocturnal activity in species that lost bright colouration and have much larger eyes than their closest relative – mtDNA distance around 1%). Analogically, we showed similar low mtDNA differentiation in Scarelus and Cautires in Malaysia (Bray and Bocak 2016, Jiruskova et al. 2019) and in Synchonnus in Australia (Kusy et al. 2018). To sum up, our claims are downgraded, alternative estimates included, and reference to further evidence added.

Also, It is surprising the amount of text used for the phylogenomic analyses, it is a very appropriate methodological part but barely mentioned in the results. Authors should improve the result/discussion of this part, describing and comparing what they produce with other studies (if any).

Thanks for the comment and we agree with you that we were too brief. We tried to improve these sections. We limited our discussion on the position of individual taxa as we wrote the manuscript for wide audience without specific knowledge on our group. As we noted in the manuscript, morphology must be inevitably studied, but it is not included in the present study. We already work on the next step of biodiversity research: the generic classification of the group with integrative definitions of named taxa. We have already prepared several hundreds of illustrations and the analysis of the congruence of relationships based on morphological traits, phylogenomics, and mitochondrial markers. But the taxon-specific information is interesting for taxonomists, not for wide audience.

Finally, the use of "person-months of focused field research" should be considered to be removed, as it did not provide any results or comparison with other study groups (which won't be possible).

Thanks for the suggestion that we fully follow in the revised manuscript. We added explanation why standardized methods were not used and we kept only approximate information which regions were sparsely sampled.

Reviewer #3 (Recommendations for the authors):Conservation efforts must be evidence-based, so rapid and economically feasible methods should be used to quantify diversity and distribution patterns. The principal objective of this study is to demonstrate how biodiversity information for a hyperdiverse tropical group can be rapidly expanded via targeted field research and large-scale sequencing. The authors have attempted to overcome current impediments to the gathering of biodiversity data by using integrative phylogenomic and three mtDNA fragment analyses. As a model, they sequenced the Metriorrhynchini beetle fauna, sampled from ~700 localities in three continents. The species-rich dataset included ~6,500 terminals, >2,300 putative species, more than a half of them unknown to science. It is an amazing finding. Their information and phylogenetic hypotheses can be a resource for higher-level phylogenetics, population genetics, phylogeographic studies, and biodiversity estimation. At the same time, they want to show how limited the taxonomical knowledge is and how this lack is hindering biodiversity research and management. It is a nice and well-designed study with a very important meaning. I would be happy to recommend it to be accepted after the revision. However before it is formally accepted, there are several questions followed:1. The definition of species is very complex. The molecular species, e.g. OTU, is still different from morphological species using traditional taxonomic methods. Of course, it is very common to use molecular species to define and infer biodiversity in many studies. For this study, ideally, the authors can express their comments on the relationships between molecular species and morphological species. For example, Sharkey et al. (2021 Zookeys) published 403 new species only with molecular information.

Thanks for an interesting suggestion. We added several sentences that summarize the relationships between nextRAD, mtDNA and morphology based species in Metriorrhynchini as we have studied them in previous works (Bocak and Yagi 2010, Kalousova and Bocak 2017, Bocek et al. 2019, Jiruskova et al. 2019). While trying to avoid selfcitations in the earlier version, we underestimated the need to support our conclusions by earlier studies. Our earlier discussion on the delimitation of species in Metriorrhynchini (especially the nextRAD,mtDNA study in Frontiers of Zoology) have shown that the net-winged beetle are often well morphologically differentiated even when species and have recently split. The studies discussed also the factors contribution to observed patterns: (the mosaic structure of mimetic communities, altitudinal differences, timing of activity etc.). The performance of delimitation must be evaluated with morphology and nuclear genes and can be solved only in restricted taxonomic studies. We consider our species numbers as estimates and do not intend to describe species without morphological evaluation. The text was modified accordingly.

2. P8L199: The constituent clades should be point out based on which tree. Pruned mitogenomic tree with and without constraints or Constrained mitogenomics?

Thanks for pointing to the problem. We expanded the text. To sum up, we used constrained mtDNA topology which contains all genomic samples in fixed position and we listed in the Supplements also genera for which transcriptomes or whole genomes have not been sequenced. The genera for which no molecular data are available are assigned to subclades using the morphology. The senior author have studied all type species of Metriorrhynchini genera and about 70 percent of all types of Metriorrhynchini.

3. P8L205-208: How many mOTUs if using thresholds of 2%? 2345 or 2356? It is inconsistent in this paragraph.

I am very sorry for this error, all number are checked and are now consistent. The wrong number was overlooked after the latest (smaller, rigorously edited) dataset was used for delimitation.

4. P10L244: This table is not too clearly to explain shared regions. If illustrate the meanings of number before and after "/", it will be better.

Thanks for your comment, we added the explanation to the headings of columns.

5. P14L369: What's the references?

The references were added. Now we cite the morphology-based phylogenetic analysis and the latest catalogue – (Bocak 2002, Bocak et al. 2020). To keep the list of references focused on relevant and more general works, we only referred readers to the history of classification described in these studies. The latest study contains almost complete bibliography for Metriorrhynchini.

6. P16L448: This discovery should be shown in RESULTS part.

Thanks for the comment, we moved illustration forward and added statements about the findings to Results.

7. P18L528: What is the voucher number of Metanoeus sp.?

The voucher number is now added.

8. P96 S18: The figure S18 is unclear. It should be replaced by clear one.

The Figure S18 is replaced by the high resolution version.